# **XYZFlow**: Scaling Multidimensional Shortcut Flows for Efficient Generative Modeling

**Jinxiu Liu** [1] [*]   **Xuanming Liu** [2] [*]   **Kangfu Mei** [3]   **Yandong Wen** [2]   **Weiyang Liu** [1]

**spherelab.ai/xyzflow**

## Abstract

High-fidelity image generation faces a trade-off between speed and quality. Diffusion models produce strong visuals but require costly iterative sampling. Existing efficient methods mainly distill pretrained models into few-step samplers, a challenging process that depends heavily on teacher-model quality. In this paper, we introduce XYZFlow, a framework that rethinks efficient generation through multidimensional scaling of flow matching. Unlike single-step mappings, XYZFlow enhances expressivity by making probability paths more identifiable and learnable through structured multidimensional conditioning. We view autoregressive modeling as implicit flow straightening, where richer context reduces trajectory ambiguity. XYZFlow realizes this idea through two orthogonal dimensions: temporal scaling, which uses non-Markovian conditioning on the full denoising history; and spatial scaling, enabled by Next Shortcut Prediction, which sequentially generates patches using preceding patches' denoising trajectories as priors. Experiments show that XYZFlow achieves state-of-the-art performance, with $7.2$-$8.5\times$ teacher speedups and competitive FID, while Next Shortcut Prediction delivers superior quality-latency trade-offs over model scaling or step reduction.

## 1. Introduction

Generative models, particularly diffusion probabilistic models, have revolutionized synthetic data generation across various modalities (Sohl-Dickstein et al., 2015; Song & Ermon, 2019; Ho et al., 2020; Rombach et al., 2022; Ho et al., 2020). The dominant paradigm involves a gradual forward process that incrementally corrupts data with noise, followed by a learned reverse process for iterative data reconstruction. While models like DDPM (Ho et al., 2020) and Score-SDE (Song et al., 2021b) achieve remarkable quality, this performance comes at a substantial computational cost (Lu et al., 2022b;a; Karras et al., 2022), often requiring hundreds of neural function evaluations per sample. Such a cost prevents these models from real-time applications.

The pursuit of efficiency has centered on a key insight: few-step generation quality fundamentally depends on the uniqueness of the noise-to-data trajectory mapping (Lipman et al., 2023; Frans et al., 2025; Boffi et al., 2024; Salimans & Ho, 2022). This uniqueness enables effective distillation by reducing the problem from learning complex distributions to fitting deterministic functions. Pioneering methods like Rectified Flows (Lipman et al., 2023), Consistency Models (Song et al., 2023) and Shortcut Models (Frans et al., 2025) address this by constructing straight, deterministic probability flows through novel training objectives. However, these approaches primarily focus on improvements to distillation algorithms themselves, which is a challenging and model-dependent endeavor (Salimans & Ho, 2022; Geng et al., 2024a; Sauer et al., 2024).

Despite recent progresses, we identify another fundamental challenge that remains largely unexplored: *how can we scale the expressive power of generative models under strict sampling step constraints, without relying solely on distillation strategies? More profoundly, can we design probability flows that are intrinsically more unique and learnable through model architecture?* As conceptually visualized in Figure 1(a), the ambiguous trajectories in conventional denoising stem from a lack of focused constraints.

In this paper, we introduce a new scaling paradigm. Instead of extensive scaling through added parameters or steps, we scale *intensively* by enhancing probability flow expressivity via structured, multidimensional conditionalization. We reinterpret autoregressive modeling not merely as a gener-

[1]CUHK [2]Westlake University [3]Johns Hopkins University. Correspondence to: Jinxiu Liu <jinxiuliu0628@foxmail.com>, Weiyang Liu <wyliu@cse.cuhk.edu.hk>.

*Proceedings of the $43^{rd}$ International Conference on Machine Learning*, Seoul, South Korea. PMLR 306, 2026. Copyright 2026 by the author(s).

Jinxiu Liu's and Xuanming Liu's work was completed during internships at CUHK and Westlake, respectively.

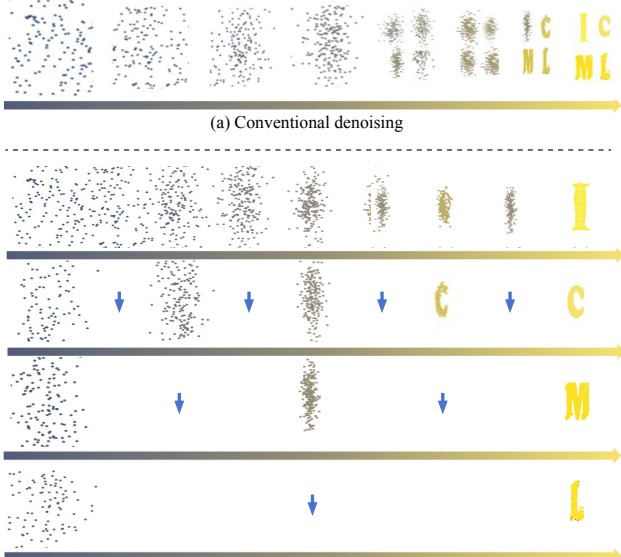

(a) Conventional denoising

(b) Our Next Shortcut Prediction paradigm

*Figure 1.* (a) Conventional one-shot denoising suffers from overlapping and ambiguous probability paths (blurred results) as the model attempts to denoise the entire image at once. (b) Our **Next Shortcut Prediction** paradigm: Denoising proceeds sequentially patch-by-patch (*e.g.*, for patches **I, C, M, L**). The **rightward small arrows** trace the denoising trajectory of each patch over time. Crucially, the **downward blue arrows** transfer the complete denoising trajectory of the preceding patch as a powerful prior. This allows subsequent patches to leverage the established context, straightening their paths and requiring fewer denoising steps (longer horizontal sequences) to achieve high fidelity.

ative strategy, but as an implicit mechanism for flow enhancement and uniqueness enforcement (Li et al., 2024). The expanding autoregressive context imposes progressively specific constraints, reducing flow variance and straightening trajectories. This perspective inherits the insight from flow straightening methods (Lipman et al., 2023; Frans et al., 2025; Albergo & Vanden-Eijnden, 2023) where deterministic paths are crucial for efficient distillation, demonstrating that *structured conditioning* enforces such uniqueness.

Guided by this insight, we propose XYZFlow, a framework that scales flow matching along two orthogonal dimensions for high-fidelity few-step generation, complementary to the prevailing path of distillation-based step compression. (1) **Temporal Scaling**: We condition each flow step on the complete history of previous states, creating a temporal coordinate system that straightens trajectories. This transforms denoising from Markovian to non-Markovian, where the KV cache of past states acts as a conditioning anchor, inspired by recent advances in recurrent diffusion processes (Hang et al., 2025). (2) **Spatial Scaling**: We propose *Next Shortcut Prediction* based on next-patch prediction. By dividing images into grids (*e.g.*, $2 \times 2$), this mechanism sequentially generates patches. Unlike standard patch-wise generation that treats patches independently, our method explicitly trans-

fers the *denoising trajectory* (not just the final output) as an effective prior for subsequent patches. As illustrated in Figure 1(b), the full denoising trajectory of the first patch serves as a conditional guidance, enabling faster generation without any quality loss.

In principle, we demonstrate that multidimensional scaling equips probability paths with high-dimensional coordinate systems. While flow straightening methods seek unique points in one-dimensional flows, our approach enhances uniqueness through orthogonal dimensional coordinates. We ground our method in two crucial principles: (1) strengthening conditions can drive reverse process variance toward zero, ensuring mapping uniqueness (Song et al., 2023; Song & Dhariwal, 2024; Geng et al., 2024b; Lu & Song, 2025; Yang et al., 2024); (2) autoregressive trajectories of preceding patches can effectively guide subsequent predictions, straightening paths in few-step regimes (Hang et al., 2025; Yan et al., 2025; Wang et al., 2024; Ren et al., 2024). Our contributions are summarized below:

- **Novel Scaling Paradigm.** We propose to scale generative models by enhancing probability flow expressivity through structured multidimensional conditionalization, advocating that scaling *constraint dimensionality* provides a principled path to mapping uniqueness.

- We introduce XYZFlow, a practical framework that implements both temporal and spatial flow scaling, featuring **Next Shortcut Prediction** for efficient inference-time cross-patch implicit trajectory guidance.

- **Strong Theoretical Justification.** We establish a theoretical framework that formalizes autoregressive modeling as explicit flow enhancement, and empirically demonstrate competitive few-step generation on ImageNet 256×256, highlighting dimensional scaling as a promising alternative to conventional approaches.

## 2. Related Work

**Few-step Diffusion and Flow Matching.** Diffusion models (Sohl-Dickstein et al., 2015; Song & Ermon, 2019; Ho et al., 2020; Song et al., 2021b;a) and their flow matching extensions (Lipman et al., 2023; Albergo & Vanden-Eijnden, 2023; Liu et al., 2022b) have established a powerful framework for generative modeling. Current research on efficient sampling primarily follows a path of *extensive scaling*, focusing on refining distillation algorithms or training objectives. Distillation-based methods (Salimans & Ho, 2022; Geng et al., 2024a; Sauer et al., 2024; Luo et al., 2024; Yin et al., 2024) aim to compress pre-trained models, while consistency-type approaches (Song et al., 2023; Song & Dhariwal, 2024; Geng et al., 2024b; Lu & Song, 2025; Yang et al., 2024) enforce self-consistency constraints along trajectories. Recent works like Flow Map Matching (Boffi

et al., 2024) and Shortcut Models (Frans et al., 2025) further explore self-consistency principles to straighten probability paths, with Inductive Moment Matching (Zhou et al., 2025) modeling the self-consistency of stochastic interpolants at different time steps. While these methods have advanced the state-of-the-art performance, their reliance on distillation algorithm improvements represents a form of extensive scaling that faces fundamental challenges in model dependency and optimization complexity. In contrast, our work addresses the core challenge of trajectory uniqueness with *intensive scaling*, which enhances flow expressivity through multidimensional conditionalization rather than pursuing better distillation strategies for existing flows.

**Autoregressive Models for Visual Generation**. Autoregressive image generation has evolved from discrete tokenization (Razavi et al., 2019; Esser et al., 2021; Lee et al., 2022) to continuous representations that avoid quantization errors (Li et al., 2024; Ren et al., 2024; 2025; Wu et al., 2025). Methods like MAR (Li et al., 2024) and DISA (Zhao et al., 2025) combine autoregressive modeling with diffusion processes, while acceleration techniques focus on caching (Yan et al., 2025) or speculative decoding (Wang et al., 2024). FAR (Hang et al., 2025) replaces the diffusion head of MAR (Li et al., 2024) with a shortcut model, accelerating through architectural changes. In a concurrent work, DART (Gu et al., 2025) and ARD (Kim et al., 2025) employ an autoregressive model that sequences 2D token maps from the diffusion process. However, unlike DART which is a standard multi-step diffusion model and only apply autoregression in denoising dimension, we reinterpret autoregressive modeling as an implicit mechanism for flow enhancement and uniqueness enforcement not only in denoising dimension but also in spatial dimension, and propose a distillation method XYZFlow that operates in the low-step regime. Specifically, XYZFlow reduces inference steps to 3-4 by following the backward ODE trajectories of a pre-trained diffusion model, while DART uses the forward noising process on ground truth data.

## 3. The Proposed XYZFlow Framework

We introduce XYZFlow, a framework that enhances the expressivity of flow models through multidimensional conditioning. Motivated by the view that autoregressive modeling provides an effective mechanism for flow straightening by incrementally imposing constraints, we propose a new training objective, Next Shortcut Prediction, which enables efficient generation via multidimensional conditioning. We interpret the expanding autoregressive context as a sequence of increasingly specific and gradually stronger constraints that reduce variance in the probability flow and straighten trajectories. This perspective extends existing insights from flow straightening methods by showing how structured conditional information can better promote path

uniqueness. Within this framework, Next Shortcut Prediction operationalizes the principle of intensive scaling, as it leverages spatial constraints to construct a high-dimensional coordinate system that effectively enforces flow uniqueness and yields straighter trajectories.

### 3.1. Autoregressive Modeling as Flow Enhancement

Traditional autoregressive approaches frame image generation as a sequence of conditional predictions. Given an image divided into patches $\langle \mathbf{x}^1, \mathbf{x}^2, \ldots, \mathbf{x}^P \rangle$, the autoregressive generation process is formulated as the chain rule:

$$p(\mathbf{x}^1, \mathbf{x}^2, \ldots, \mathbf{x}^P) = \prod_{p=1}^{P} p(\mathbf{x}^p \mid \mathbf{x}^1, \ldots, \mathbf{x}^{p-1}). \quad (1)$$

While this formulation is mathematically sound, we reconceptualize it through the lens of flow enhancement. The growing context $\mathbf{x}^1, \ldots, \mathbf{x}^{p-1}$ acts as a set of progressively stronger constraints, which can gradually reduce the variance of the conditional distribution $p(\mathbf{x}^p | \cdots)$ and straightens the probability flow path from noise to data. This conceptual shift allows us to leverage autoregressive structure not just for sequential prediction, but for intrinsically making the flow more unique and deterministic. This insight has been used for ill-posed 3D reconstruction (Liu et al., 2022a; Wen et al., 2021) and video generation (Liu et al., 2026).

Formally, we define flow enhancement as the process where conditional information $C$ transforms a base probability flow $p(\mathbf{x})$ into a conditioned flow $p(\mathbf{x}|C)$ with reduced path variance: $\mathbb{V}[\mathbf{x}_t|C] < \mathbb{V}[\mathbf{x}_t]$, leading to straighter and more deterministic trajectories. This variance reduction directly contributes to mapping uniqueness. For further theoretical justification, please refer to the Appendix.

### 3.2. A Motivating Observation from Progressive Constraint Strengthening

Our approach is motivated by the empirical observation that as more patches are generated, the conditional distribution becomes more constrained, making subsequent patches easier to sample. As illustrated in Figure 2, this observation manifests in three important phenomena:

**Next patches can be better predicted at later generation stages.** When we probe the conditional strength by predicting $\mathbf{x}^p$ based on the representation of previously generated patches, predictions for early positions are blurry and lack detail, while predictions for later positions become increasingly precise. This demonstrates that accumulated context provides stronger conditional guidance, as shown in the panda image sequence (top-right of Figure 2).

**The variance of latent patch distributions decreases for later patches.** When sampling multiple possible patches for each position during generation, the variance among

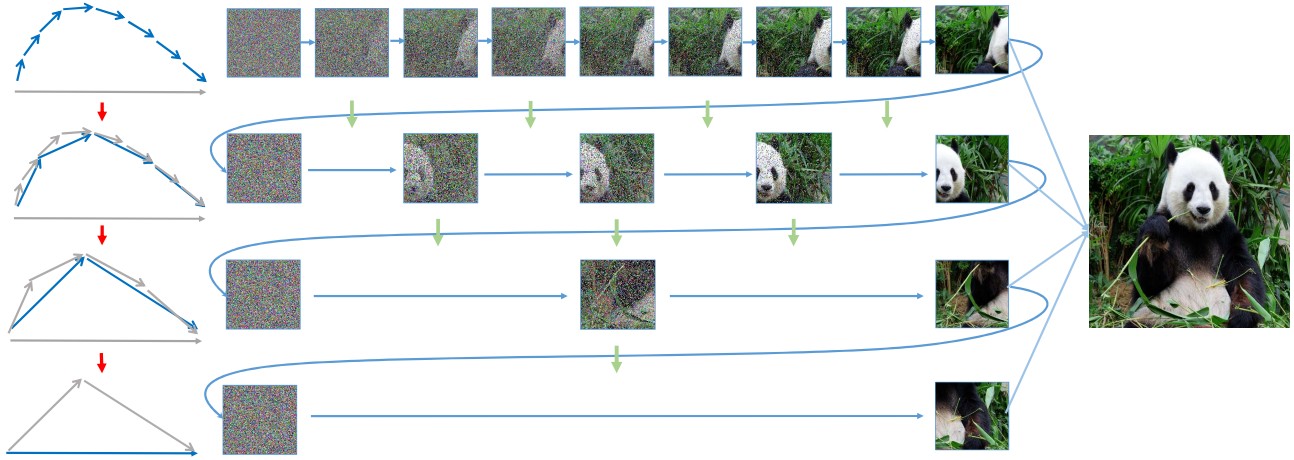

*Figure 2.* **Next Shortcut Prediction in XYZFlow**. (Top-Left) Flow diagram showing the generation sequence, where a blue curve represents progressively strengthening constraints. (Top-Right) Visualization of a non-uniform patch-based denoising process: the first image patch undergoes the most denoising steps, while subsequent patches are generated with fewer steps ("shortcuts"). This forms a long autoregressive sequence where the denoising flow from prior patches (green and blue arrows) guides the denoising of subsequent ones.

sampled patches is high for early positions but decreases significantly for later positions, indicating a more concentrated and simpler distribution under stronger conditioning. This is visually validated by the progression from noisy to clean patches in the bottom rows of Figure 2.

**Denoising paths become straighter for later patches.** Following Rectified Flow, we measure path straightness using:

$$S(\{\mathbf{x}_t\}_{t=0}^1, \mathbf{z}) = \mathbb{E}_{t\sim[0,1]}\left[\|(\mathbf{x}_1 - \mathbf{x}_0) - \mathbf{v}_\theta(\mathbf{x}_t \mid t, \mathbf{z})\|^2\right]. \quad (2)$$

Our experiments have demonstrated that $S$ decreases for patches generated later in the sequence, validating that strong contextual conditioning can effectively straighten the flow. The blue curve in Figure 2 (on the left) illustrates this path straightening effect.

### 3.3. Multidimensional Conditioning

Building on these observations, XYZFlow implements a dual-path conditioning architecture that enhances the probability flow along both **temporal** and **spatial** dimensions through Next Shortcut Prediction.

**Temporal Conditioning: Intra-patch Trajectory Conditioning**. To strengthen the conditioning along the temporal dimension, we propose to enhance the flow matching process for each patch by conditioning it on its own generation history. Specifically, for a patch $\mathbf{x}^p$, the conditioning signal at time $t$ is its entire state trajectory from the beginning of generation up to, but not including, the current time $t$. We denote this generation history as $\mathcal{H}_t^p = \{\mathbf{x}_\tau^p\}_{\tau=0}^{t-\Delta t}$. The temporal conditioning loss for the patch $\mathbf{x}^p$ is defined as the deviation of the predicted flow from the true conditional flow, given this patch's historical context $\mathcal{H}_t^p$. Therefore,

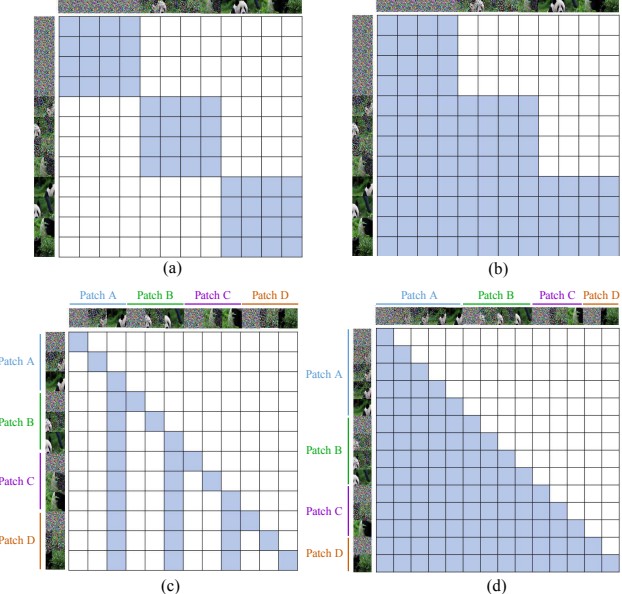

*Figure 3.* Illustration of attention mechanisms for image generation. (a) Vanilla Image Generation: Standard full-image denoising with independent patch processing. (b) Autoregressive in Denoising Dimension: Sequential denoising across patches over time. (c) Next Patch Prediction: Complete denoising of one patch before starting the next. (d) Next Shortcut Prediction: Early patches undergo more denoising steps, with full denoising trajectories of previous patches conditioning subsequent ones.

the loss can be formulated as follows:

$$\mathcal{L}_{\text{temp}}^p = \mathbb{E}_{t,\mathbf{x}_0^p,\mathbf{x}_1^p}\|v_\theta(\mathbf{x}_t^p|t,\mathcal{H}_t^p) - (\mathbf{x}_1^p - \mathbf{x}_0^p)\|^2, \quad (3)$$

where this self-conditioning with self attention can act as a strong generative prior, stabilizing the generation path by providing a temporal coordinate system for the flow.

**Spatial Conditioning: Inter-patch Trajectory Conditioning.** In the spatial dimension, the generation of each patch is conditioned on the complete trajectories of all previously generated patches. Unlike intra-patch temporal conditioning, which operates within the history of a single patch, spatial conditioning captures dependencies across patches. As illustrated in Figure 3, the key innovation is that each patch depends not only on the final content of preceding patches, but also on their full generation trajectories. This provides a substantially richer contextual signal, thereby enhancing flow expressivity across the spatial domain:

$$p(\mathbf{x}^p|\mathbf{x}^1,\ldots,\mathbf{x}^{p-1}) = p(\mathbf{x}^p|\mathcal{T}_{<p}) \quad (4)$$

where $\mathcal{T}_{<p} = \{\tau^1, \tau^2, \ldots, \tau^{p-1}\}$ and $\tau^i = \{\mathbf{x}_t^i\}_{t=0}^1$ represents the complete generation trajectory of patch $i$. Conditioning on full trajectories $\mathcal{T}_{<p}$ rather than just final states provides significantly stronger constraints: each trajectory $\tau^i$ adds multiple temporal anchor points that collectively reduce the solution space for generating $\mathbf{x}^p$, making the reverse process more deterministic. The attention patterns shown in Figure 3 demonstrate how different attention mechanisms can effectively integrate trajectory condition.

### 3.4. Next Shortcut Prediction

The core contribution of XYZFlow is Next Shortcut Prediction, which shifts the scaling paradigm from resource-intensive scaling to constraint-intensive scaling. Rather than increasing model size or the number of training steps, we scale the dimensionality of the conditioning constraints by training the model to generate effectively under progressively stronger constraints from $\mathcal{T}_{<p}$. As illustrated in Figure 2, Next Shortcut Prediction trains the model to exploit rich contextual information for accelerating the generation process. Specifically, we define a progressive denoising schedule where each patch $p$ is assigned fewer denoising steps as it is generated later in the sequence:

$$T(p) = T_{\text{full}} - \Delta T \cdot (p-1) \quad \text{for} \quad p = 1, \ldots, P, \quad (5)$$

with the constraint $T(p) \geq T_{\min} > 0$. Our training objective is formulated as teacher model trajectory distillation. The key insight is to enhance the uniqueness and straighten the probability flow by imposing powerful, structured constraints. We achieve this with an autoregressive formulation:

$$p(\mathbf{x}_0^p \mid \mathcal{T}_{<p}) = p_{\text{prior}}(\mathbf{x}_{T(p)}^p) \times \prod_{t=1}^{T(p)} p(\mathbf{x}_{t-1}^p \mid \mathbf{x}_{T(p):t}^p, \mathcal{T}_{<p}), \quad (6)$$

where $\mathbf{x}_{T(p):t}^p = [\mathbf{x}_{T(p)}^p, \mathbf{x}_{T(p)-1}^p, \ldots, \mathbf{x}_t^p]$ denotes the historical denoising trajectory. This formulation offers two fundamental advantages that embody our principle of intensive scaling. First, it equips each denoising step with a higher-dimensional coordinate system for conditioning.

Specifically, the combination of spatial context from previously generated patches, $\mathcal{T}_{<p}$, and temporal context from the historical trajectory, $\mathbf{x}_{T(p):t}^p$, imposes a highly specific constraint. This drastically reduces the variance of the reverse process, transforming the mapping from $\mathbf{x}_t^p$ to $\mathbf{x}_{t-1}^p$ from an ambiguous, one-to-many problem into a nearly deterministic, one-to-one function, thereby straightening the probability flow path. Second, it enables synergistic information fusion. To predict $\mathbf{x}_{t-1}^p$ at every step, the model learns to integrate both coarse-grained and fine-grained information. The recent denoised sample $\mathbf{x}_t^p$ is the best source for fine-grained details, while the historical trajectory closer to $\mathbf{x}_{T(p)}^p$ provides better coarse-grained structural information. We aim to estimate $p(\mathbf{x}_{t-1}^p \mid \mathbf{x}_{T(p):t}^p, \mathcal{T}_{<p})$, which, under our strong conditioning, approximates a Dirac delta distribution. This is achieved within the Flow Matching framework by defining the mapping function:

$$\begin{aligned} \mathbf{x}_{t-1}^p &= G(\mathbf{x}_{T(p):t}^p, \mathcal{T}_{<p}, t) \\ &:= \mathbf{x}_t^p + (\gamma(t-1) - \gamma(t)) \cdot v_\theta(\mathbf{x}_{T(p):t}^p, t, \mathcal{T}_{<p}), \end{aligned} \quad (7)$$

which is approximated by our student neural network $v_\theta$ using an Euler step. Here, $\gamma$ is the noise schedule. The complete training objective integrates multidimensional conditioning with this progressive schedule. It distills the teacher's trajectory by regressing the target sample:

$$\mathcal{L}_{\text{NextShortcut}} = \mathbb{E}_{p\sim[1,P]}\left[\sum_{t=1}^{T(p)} \left\| G_\theta(\mathbf{x}_{T(p):t}^p, t, \mathcal{T}_{<p}) - \mathbf{x}_{t-1}^p \right\|_2^2\right]. \quad (8)$$

Here we have that $G_\theta(\mathbf{x}_{T(p):t}^p, t, \mathcal{T}_{<p}) = \mathbf{x}_t^p + (\gamma(t-1) - \gamma(t)) \cdot v_\theta(\mathbf{x}_{T(p):t}^p, t, \mathcal{T}_{<p})$ represents the student's one-step prediction. The transformer architecture allows computing $G_\theta$ for all $t$ simultaneously by using an attention mask. We design the attention mask to be block-wise causal, allowing the model to use the entire trajectory history $\mathbf{x}_{T(p):t}^p$ as context, which is the most flexible and effective option. This objective directly embodies our intensive scaling principle: it trains the student network to predict the optimal denoising path using both temporal (historical trajectory) and spatial (previous patches) conditioning. The green arrows in Figure 2 illustrate this accelerated generation path. Our framework can also benefit from an **additional discriminator loss** applied to the final generated patch $\hat{\mathbf{x}}_0^p$. This adversarial training, which uses real data as supervision, further enhances the high-frequency details in the generated outputs. During inference, the model generates the first patch with the full step budget $T(1) = T_{\text{full}}$ to establish a robust and reliable anchor. Each subsequent patch $p \geq 2$ is generated in an autoregressive manner. Starting from $\mathbf{x}_{T(p)}^p \sim p_{\text{prior}}$, at each step $t$, the model predicts $\hat{\mathbf{x}}_{t-1}^p = G_\theta(\hat{\mathbf{x}}_{T(p):t}^p, t, \mathcal{T}_{<p})$ based on the entire history of predictions $\hat{\mathbf{x}}_{T(p):t}^p = [\mathbf{x}_{T(p)}^p, \hat{\mathbf{x}}_{T(p)-1}^p, \ldots, \hat{\mathbf{x}}_t^p]$. The information of the historical predictions is efficiently managed

| Model | #params | AR steps | Diff steps | FID↓ | IS↑ | Pre.↑ | Rec.↑ | Time (s)↓ | Speed-Up↑ |
|---|---|---|---|---|---|---|---|---|---|
| *Base Models (170M-208M parameters)* | | | | | | | | | |
| MAR-B (Li et al., 2024) | 208M | 256 | 100 | 2.31 | 281.7 | 0.82 | 0.57 | 0.650 | 1.0× |
|  |  | 64 | 50 | 2.39↑0.08 | 281.0↓0.7 | 0.82 | 0.57 | 0.134↓0.516 | 4.9×↑3.9 |
| FlowAR-S (Ren et al., 2024) | 170M | 5 | 25 | 3.70↑1.39 | 235.1↓46.6 | 0.81↓0.01 | 0.51↓0.06 | 0.024↓0.626 | 27.1×↑26.1 |
| xAR-B (Ren et al., 2025) | 172M | 4 | 50 | 1.67↓0.64 | 265.2↓16.5 | 0.80↓0.02 | 0.62↑0.05 | 0.130↓0.520 | 5.0×↑4.0 |
| XYZFlow-B (w/o GAN) | 172M | 4 | 5→2 | 2.02↓0.29 | 261.1↓20.6 | 0.80↓0.02 | 0.58↑0.01 | 0.018↓0.632 | 36.1×↑35.1 |
| XYZFlow-B (w/ GAN) | 172M | 4 | 5→2 | 1.63↓0.68 | 268.5↓13.2 | 0.81↓0.01 | 0.62↑0.05 | 0.018↓0.632 | 36.1×↑35.1 |
| *Large Models (479M-820M parameters)* | | | | | | | | | |
| DiT/XL-2 (Peebles & Xie, 2023) | 676M | - | 25 | 2.89 | 230.2 | 0.80 | 0.57 | 0.494 | 1.0× |
| DART | 812M | - | 16 | 5.62↑2.73 | 231.7↑1.5 | 0.78↓0.02 | 0.55↓0.02 | 0.075↓0.419 | 6.6×↑5.6 |
| DART-AR (Gu et al., 2025) | 812M | 4096 | - | 3.98↑1.09 | 256.8↑26.6 | 0.80 | 0.58↑0.01 | 7.44↑6.946 | 0.07×↓0.93 |
| DART-FM | 820M | - | 16 | 3.82↑0.93 | 263.8↑33.6 | 0.81↑0.01 | 0.60↑0.03 | 0.32↓0.174 | 1.5×↑0.5 |
| MeanFlow-XL/2 (w/o CFG) | 676M | - | 1 | 3.43↑0.54 | - | - | - | 0.009↓0.485 | 54.9×↑53.9 |
| MeanFlow-XL/2 | 676M | - | 1 | 2.93↑0.04 | - | - | - | 0.018↓0.476 | 27.4×↑26.4 |
| MeanFlow-XL/2+ | 676M | - | 1 | 2.20↓0.69 | - | - | - | 0.018↓0.476 | 27.4×↑26.4 |
| DART Distill | 676M | - | 2 | 10.92↑8.03 | 167.0↓63.12 | 0.68↓0.12 | 0.52↓0.05 | 0.033↓0.461 | 15.0×↑14.0 |
| ARD (w/o GAN) | 676M | - | 2 | 6.29↑3.40 | 188.0↓42.15 | 0.74↓0.06 | 0.56↓0.01 | 0.034↓0.460 | 14.5×↑13.5 |
| DART Distill (w/o GAN) | 676M | - | 4 | 10.25↑7.36 | 181.6↓48.62 | 0.70↓0.10 | 0.47↓0.10 | 0.065↓0.429 | 7.6×↑6.6 |
| ARD (w/o GAN) | 676M | - | 4 | 4.32↑1.43 | 209.0↓21.2 | 0.77↓0.03 | 0.57 | 0.066↓0.428 | 7.5×↑6.5 |
| DART Distill (w/ GAN) | 676M | - | 4 | 3.84↑0.95 | 221.1↓9.1 | 0.78↓0.02 | 0.55↓0.02 | 0.065↓0.429 | 7.6×↑6.6 |
| ARD (w/ GAN) | 676M | - | 4 | 1.84↓1.05 | 235.8↓5.6 | 0.80 | 0.62↑0.05 | 0.066↓0.428 | 7.5×↑6.5 |
| MAR-L (Li et al., 2024) | 479M | 256 | 100 | 1.78↓1.11 | 296.0↑65.8 | 0.81↑0.01 | 0.60↑0.03 | 1.102↑0.608 | 0.4×↓0.6 |
|  |  | 64 | 50 | 1.86↓1.03 | 294.0↑63.8 | 0.80 | 0.61↑0.04 | 0.250↓0.244 | 2.0×↑1.0 |
| FlowAR-L (Ren et al., 2024) | 589M | 5 | 25 | 1.87↓1.02 | 273.1↑42.9 | 0.80 | 0.62↑0.05 | 0.124↓0.370 | 4.0×↑3.0 |
| xAR-L (Ren et al., 2025) | 608M | 4 | 50 | 1.28↓1.61 | 292.5↑62.3 | 0.82↑0.02 | 0.62↑0.05 | 0.394↑0.100 | 1.3×↑0.3 |
| XYZFlow-L (w/o GAN) | 608M | 4 | 5→2 | 1.79↓1.10 | 265.2↓35.0 | 0.81↑0.01 | 0.61↑0.04 | 0.050↓0.444 | 9.9×↑8.9 |
| XYZFlow-L (w/ GAN) | 608M | 4 | 5→2 | 1.25↓1.64 | 295.8↑65.6 | 0.83↑0.03 | 0.63↑0.06 | 0.050↓0.444 | 9.9×↑8.9 |
| *Huge Models (943M-2.0B parameters)* | | | | | | | | | |
| FlowAR-H (Ren et al., 2024) | 1.9B | 5 | 50 | 1.67 | 276.3 | 0.80 | 0.62 | 0.423 | 1.0× |
| VAR-d30 (Tian et al., 2024) | 2.0B | 10 | - | 1.92↑0.25 | 323.1↑46.8 | 0.82↑0.02 | 0.59↓0.03 | 0.039↓0.384 | 10.8×↑9.8 |
| MAR-H (Li et al., 2024) | 943M | 256 | 100 | 1.55↓0.12 | 303.7↑27.4 | 0.81↑0.01 | 0.62 | 1.957↑1.534 | 0.2×↓0.8 |
|  |  | 64 | 50 | 1.65↓0.02 | 299.8↑23.5 | 0.80 | 0.62 | 0.462↑0.039 | 0.9×↓0.1 |
| xAR-H (Ren et al., 2025) | 1.1B | 4 | 50 | 1.24↓0.43 | 301.6↑25.3 | 0.83↑0.03 | 0.64↑0.02 | 0.896↑0.473 | 0.5×↓0.5 |
| XYZFlow-H (w/o GAN) | 1.1B | 4 | 5→2 | 1.73↓0.06 | 271.5↓4.8 | 0.82↑0.02 | 0.62 | 0.105↓0.318 | 4.0×↑3.0 |
| XYZFlow-H (w/ GAN) | 1.1B | 4 | 5→2 | 1.22↓0.45 | 304.2↑27.9 | 0.84↑0.04 | 0.64↑0.02 | 0.105↓0.318 | 4.0×↑3.0 |

*Table 1.* **Main results of different methods** on ImageNet 256×256. Models are organized by parameter count from small to large. Colored numbers indicate performance change relative to baseline models.

using a key-value cache. This process leverages the accumulated context $\mathcal{T}_{<p}$ and the learned ability to exploit constraints, achieving significant speedup while maintaining generation quality through constraint exploitation.

# 4. Experiments and Results

We empirically validate the efficacy of XYZFlow, focusing on the theoretical claims in Section 3. Our experiments demonstrate that: (1) Multidimensional conditioning straightens the probability flow for subsequent patches, enabling better generation quality; (2) The Next Shortcut Prediction objective effectively trains models to utilize accumulated context for accelerated generation by decreasing total denoising steps; (3) XYZFlow achieves promising efficiency-quality trade-offs in image synthesis.

## 4.1. Experimental Setup

We train and evaluate XYZFlow on the ImageNet 256×256 class-conditional generation benchmark (Deng et al., 2009).

Training is conducted on 8 NVIDIA H100 GPUs for 300K steps with a batch size of 128 and a learning rate of 0.0001. To better evaluate scalability, we employ three autoregressive teacher models of varying sizes, including Base (172M), Large (608M), and Huge (1.1B) (Ren et al., 2025). ODE trajectory data is generated by running each teacher for 50 steps with a classifier-free guidance scale of 2.3, pre-computing 2.5M trajectories for distillation. We comprehensively evaluate sample quality using four established metrics: Fréchet Inception Distance (FID) (Heusel et al., 2017), Inception Score (IS) (Salimans et al., 2016), and Precision/Recall (Dhariwal & Nichol, 2021) to quantify fidelity and diversity. Inference time (shown in seconds) and speed-up relative to baseline models are reported to measure efficiency.

## 4.2. Main Results and Analysis

Table 1 presents a comprehensive system-level comparison on ImageNet 256×256. XYZFlow achieves superior efficiency-quality trade-offs across all model scales, demonstrating the advantage of our spatio-temporal autoregressive

framework. Our key innovation lies in a unified framework for spatio-temporal autoregression: (1) Temporally, the method models the denoising trajectory, making it straighter and more predictable; (2) Spatially, it decomposes the image into a sequence of patches and predicts each subsequent patch conditioned on previous patches and the learned temporal trajectory. The linearization of the temporal trajectory provides stable and simplified global context, which significantly eases the generation task for each individual patch. This enables the use of a lightweight network for fast inference. The results validate this design. Our base model, XYZFlow-B, with only 172M parameters, matches the inference speed of the 676M-parameter one-step model MeanFlow-XL/2+ (Geng et al., 2025) (both at 0.018s per image) while achieving superior FID (1.63 vs. 2.20). Furthermore, XYZFlow-B significantly outperforms the 820M-parameter DART-FM (Gu et al., 2025) in both FID (1.63 vs. 3.82) and inference speed (0.018s vs. 0.32s). The consistent improvements across scales—XYZFlow-L (608M) and XYZFlow-H (1.1B) achieve FIDs of 1.25 and 1.22, respectively—demonstrate the scalability and effectiveness of our approach. Compared to teacher models, XYZFlow provides substantial speed-up while maintaining quality. It also outperforms other accelerated iterative methods. For instance, XYZFlow-B provides a $36.1\times$ speed-up over its teacher with an FID of 1.63, whereas FlowAR-S provides a $27.1\times$ speed-up with a higher FID of 3.70. This demonstrates that intensive modeling of the generative probability flow ("depth scaling") offers a viable and efficient alternative to simply enlarging model scale ("width scaling") or adding distillation steps. In summary, XYZFlow successfully straightens the generative trajectory through temporal conditioning, simplifies patch-wise generation via spatial decomposition, and leverages their synergy for efficient, high-fidelity image synthesis. While ARD and DART Distill variants demonstrate competitive performance at larger scales, they primarily serve to highlight the efficiency of our architectural design. For instance, ARD (w/ GAN, 676M) achieves a strong FID of 1.84 at 0.066s, and DART Distill (w/ GAN, 676M) attains an FID of 3.84 at 0.065s. However, their inference speeds are $3.6\times$ slower than our 172M-parameter XYZFlow-B (0.018s) which also achieves a better FID of 1.63. This indicates that while these methods leverage distillation and GAN augmentation for quality improvements, their underlying autoregressive or iterative structures do not fundamentally accelerate inference. In contrast, our spatio-temporal autoregressive framework fundamentally rethinks the generative process, achieving superior speed and quality with a significantly more compact model.

To further evaluate robustness under a weaker teacher, we additionally replace the xAR teacher with DiT/XL-2 (25-step) and perform patch-wise trajectory distillation under the same 256×256 setting. The results in Table 2 show that standard 5-step distillation degrades sharply with this teacher,

| Method | Params | Steps | FID↓ | Total |
|---|---|---|---|---|
| DiT/XL-2 (Teacher) | 676M | 25 | 2.89 | 25 |
| DiT/XL-2 Distilled | 676M | 5 | 8.97 | 5 |
| DiT/XL-2 Distilled (GAN) | 676M | 5 | 3.37↓5.60 | 5 |
| XYZFlow-B (Constant) | 172M | 5→5→5→5 | 3.83↓5.14 | 20 |
| XYZFlow-B | 172M | 5→4→3→2 | 3.85↓5.12 | 14 |
| XYZFlow-B (GAN) | 172M | 5→4→3→2 | **1.74↓1.63** | 14 |

*Table 2.* Weak-teacher distillation results on ImageNet 256×256 using DiT/XL-2 (25-step) as the teacher.

| Teacher | Teacher FID↓ | XYZFlow-L↓ | XYZFlow-L (GAN)↓ |
|---|---|---|---|
| xAR-B | 1.61 | 1.91↑0.30 | 1.32↓0.59 |
| xAR-L | 1.28 | 1.79↑0.51 | 1.25↓0.54 |
| xAR-H | 1.24 | 1.77↑0.53 | **1.25↓0.52** |

*Table 3.* Comparison across xAR teachers on ImageNet 256×256.

yielding FID 8.97 without GAN and 3.37 with GAN. In contrast, XYZFlow-B maintains substantially stronger performance: the progressive 5→4→3→2 schedule achieves FID 3.85 without GAN, and further improves to 1.74 with GAN. Notably, the progressive schedule matches the constant 5→5→5→5 variant (3.85 vs. 3.83) while requiring fewer total denoising steps, confirming that Next Shortcut Prediction preserves fidelity while improving efficiency. These results show that the gains of XYZFlow arise from the proposed spatio-temporal architecture itself rather than depending on a particularly strong teacher.

This trend is consistent across teachers of different strengths. As summarized in Table 3, XYZFlow-L without GAN remains stable when distilling from xAR-B, xAR-L, and xAR-H, varying only from 1.91 to 1.77 FID. After GAN fine-tuning, the gap narrows further, reaching 1.32, 1.25, and 1.25, respectively. The largest gain appears for the weakest teacher, where GAN improves XYZFlow-L by 0.59 FID, while the stronger teachers show smaller but still consistent gains of 0.54 and 0.52. This result indicates that stronger teachers are helpful, but the proposed architecture preserves robust performance even when the teacher quality degrades.

### 4.3. Ablation Study

**Component Importance Analysis**. Table 4 presents a systematic evaluation of XYZFlow's core components. Our analysis reveals the distinct contributions of each component through controlled ablations: (1) **Full History Guidance** emerges as the most critical component. Removing full history guidance ($\mathcal{T}_{<p}$) causes FID to degrade by approximately 0.5 across all model sizes (*e.g.*, 2.02→3.51 for Base). These results show that inter-patch trajectory conditioning is essential for maintaining generation quality, and also validate our hypothesis that complete trajectory information can provide richer contextual signals than final patch content alone. (2) **Shortcut Prediction** brings an interesting advantage in denoising efficiency. While having minimal impact on final quality (*e.g.*, FID differences < 0.03), it provides substantial acceleration benefits. We can observe that the "-

| Method | Params | Steps | FID↓ | IS↑ | Pre↑ | Rec↑ | Total |
|---|---|---|---|---|---|---|---|
| Teacher-Base | 172M | 50 | 1.72 | 280.4 | 0.82 | 0.59 | 200 |
| Distilled-Base | 172M | 5 | 3.03 | 225.3 | 0.78 | 0.55 | 20 |
| + Local History | 172M | 5→2 | 2.25↓0.78 | 249.8↑24.5 | 0.78 | 0.54↓0.01 | 14 |
| - Full History | 172M | 5→2 | 3.51↑0.48 | 219.9↓5.4 | 0.77↓0.01 | 0.52↓0.03 | 14 |
| - Shortcut | 172M | 5 | 2.05↓0.98 | 258.5↑33.2 | 0.80↑0.02 | 0.58↑0.03 | 20 |
| XYZFlow-B | 172M | 5→2 | 2.02↓1.01 | 261.1↑35.8 | 0.80↑0.02 | 0.58↑0.03 | 14 |
| XYZFlow-B (GAN) | 172M | 5→2 | **1.63↓1.40** | **268.5↑43.2** | **0.81↑0.03** | **0.62↑0.07** | 14 |
| Teacher-Large | 608M | 50 | 1.28 | 292.5 | 0.82 | 0.62 | 200 |
| Distilled-Large | 608M | 5 | 2.85 | 235.1 | 0.79 | 0.57 | 20 |
| + Local History | 608M | 5→2 | 2.02↓0.83 | 254.3↑19.2 | 0.79 | 0.57 | 14 |
| - Full History | 608M | 5→2 | 3.35↑0.50 | 229.3↓5.8 | 0.78↓0.01 | 0.54↓0.03 | 14 |
| - Shortcut | 608M | 5 | 1.82↓1.03 | 263.8↑28.7 | 0.81↑0.02 | 0.61↑0.04 | 20 |
| XYZFlow-L | 608M | 5→2 | 1.79↓1.06 | 265.2↑30.1 | 0.81↑0.02 | 0.61↑0.04 | 14 |
| XYZFlow-L (GAN) | 608M | 5→2 | **1.25↓1.60** | **295.8↑60.7** | **0.83↑0.04** | **0.63↑0.06** | 14 |
| Teacher-Huge | 1.1B | 50 | 1.24 | 301.6 | 0.83 | 0.64 | 200 |
| Distilled-Huge | 1.1B | 5 | 2.75 | 240.8 | 0.80 | 0.59 | 20 |
| + Local History | 1.1B | 5→2 | 1.96↓0.79 | 259.1↑18.3 | 0.80 | 0.57↓0.02 | 14 |
| - Full History | 1.1B | 5→2 | 3.25↑0.50 | 234.6↓6.2 | 0.79↓0.01 | 0.56↓0.03 | 14 |
| - Shortcut | 1.1B | 5 | 1.76↓0.99 | 268.2↑27.4 | 0.82↑0.02 | 0.61↑0.02 | 20 |
| XYZFlow-H | 1.1B | 5→2 | 1.73↓1.02 | 271.5↑30.7 | 0.82↑0.02 | 0.62↑0.03 | 14 |
| XYZFlow-H (GAN) | 1.1B | 5→2 | **1.22↓1.53** | **304.2↑63.4** | **0.84↑0.04** | **0.64↑0.05** | 14 |

*Table 4.* Ablation study of XYZFlow components. FID↓ is lower-better; IS↑, Pre↑, Rec↑ are higher-better. Total Steps↓ represents the cumulative inference steps. Colored numbers indicate performance change relative to baseline (Distilled) models. In the table, '+' and '-' denote the baseline model with and without the corresponding component, respectively.

| Cell Size | Grid Size | FID↓ | IS↑ |
|---|---|---|---|
| $1 \times 1$ | $16 \times 16$ | 2.85 | 280.7 |
| $2 \times 2$ | $8 \times 8$ | 2.46 | 283.9 |
| $4 \times 4$ | $4 \times 4$ | 2.11 | 289.2 |
| **$8 \times 8$** | **$2 \times 2$** | **2.02** | **301.5** |
| $16 \times 16$ | $1 \times 1$ | 2.55 | 283.8 |

*Table 5.* Ablation on the cell size ($k \times k$ tokens).

Shortcut" variant maintains similar FID scores but requires 20 total steps compared to XYZFlow's 14 steps. This result confirms that Shortcut Prediction primarily improves efficiency rather than quality, consistent with its design goal of exploiting straightened trajectories for faster convergence. (3) **Local History Conditioning** ($\mathcal{H}_t^p$) contributes moderately to performance, with its removal causing FID degradation of approximately 0.2-0.3. This result suggests that while intra-patch temporal conditioning provides useful stabilization, the inter-patch spatial conditioning plays a more significant role in the overall framework. (4) **Adversarial loss** consistently improves all metrics across different model sizes, which demonstrates that XYZFlow's straightened paths provide a favorable foundation for adversarial training on real data, enabling the student model to potentially exceed the teacher's capabilities.

**Ablation Study on Cell Size**. A prediction cell is formed by grouping a cluster of $k \times k$ spatially adjacent tokens. Using a larger cell size incorporates more tokens in a single prediction step, thereby capturing broader contextual information. For a $256 \times 256$ input image, the encoded continuous latent representation has a spatial resolution of $16 \times 16$. Consequently, the image can be partitioned into an $m \times m$ grid,

where each cell consists of $k \times k$ neighboring tokens. Under the XYZFlow framework, we employ a unified denoising schedule of 5 steps per patch and use the Base Model configuration. As shown in Table 5, we evaluate different cell sizes with $k \in \{1, 2, 4, 8, 16\}$. Here, $k = 1$ corresponds to a single token, while $k = 16$ represents the entire image as a single entity. Performance improves as $k$ increases, reaching its peak with an FID of 2.02 at a cell size of $8 \times 8$ (*i.e.*, $k = 8$). Beyond this point, performance declines, yielding an FID of 2.55 when the whole image is treated as one entity. These results indicate that using cells as the prediction unit, rather than the entire image, allows the model to explicitly condition on previously generated context. This conditioning enhances prediction confidence while preserving both rich semantic information and fine-grained local details.

**Analysis of Next Shortcut Prediction**. Our ablation study of shortcut prediction strategies, summarized in Table 6, yields four key insights that support our design choices. (1) **Progressive denoising step reduction achieves optimal efficiency-quality trade-off**. Our proposed 5→4→3→2 strategy achieves nearly identical quality to the constant 5-step approach (FID 1.63 vs. 1.63) but with 30% fewer total steps (14 vs. 20), demonstrating that sampling can be accelerated more aggressively in later stages without compromising quality. (2) **Initial step configuration is critical**. The comparable performance of constant strategies (5→5→5→5 vs. 4→4→4→4) highlights that an initial step of T(1)=5 (a divisor of the teacher's 50-step trajectory) provides the optimal starting point for effective distillation. (3) **Aggressive reduction can harm generation diversity**.

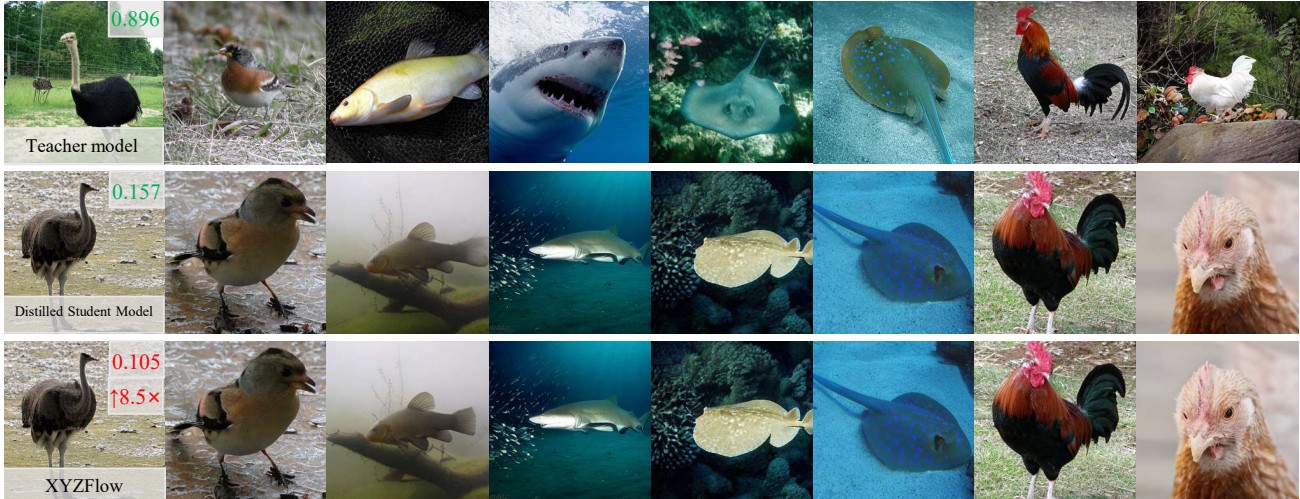

*Figure 4.* Visual comparison demonstrating the efficiency of XYZFlow. Our 1.1B-parameter model achieves an **8.5× faster** generation time than the teacher model and an additional **1.5× speedup** over the base student distillation, with no perceptible loss in quality.

| Schedule $T(p)$ | FID↓ | IS↑ | Pre↑ | Rec↑ | Total Steps↓ |
|---|---|---|---|---|---|
| Teacher (50 steps) | 1.72 | 280.4 | 0.82 | 0.59 | 200 |
| 5→4→3→2 (Ours) | 1.63↓0.09 | 268.5↓11.9 | **0.81**↓0.01 | **0.62**↑0.03 | 14↓186 |
| 8→4→2→1 (Uniform) | 1.75↑0.03 | 255.2↓25.2 | 0.77↓0.05 | 0.57↓0.02 | 15↓185 |
| 4→4→4→4 | 1.84↑0.12 | 248.9↓31.5 | 0.74↓0.08 | 0.55↓0.04 | 16↓184 |
| 4→3→2→1 | 1.88↑0.16 | 245.3↓35.1 | 0.73↓0.09 | 0.54↓0.05 | 10↓190 |
| 8→8→8→8 | **1.61**↓0.11 | **269.0**↓11.4 | **0.81**↓0.01 | **0.62**↑0.03 | 32↓168 |
| 5→5→5→5 (Constant) | 1.63↓0.09 | 267.9↓12.5 | **0.81**↓0.01 | 0.61↑0.02 | 20↓180 |
| 5→4→4→2 | 1.64↓0.08 | 266.2↓14.2 | 0.80↓0.02 | 0.60↑0.01 | 15↓185 |
| 5→2→2→2 (Aggressive) | 1.70↓0.02 | 258.6↓21.8 | 0.78↓0.04 | 0.58↓0.01 | 11↓189 |

*Table 6.* Ablation study of Next Shortcut Prediction strategies on Base Model (172M). FID↓ is lower-better; IS↑, Pre↑, Rec↑ are higher-better. Total Steps↓ represents the cumulative inference steps. Colored numbers indicate performance change relative to baseline (50 steps). **Bold** indicates best performance, underline indicates second best.

The 5→2→2→2 strategy shows degraded recall (0.58 vs. 0.62), confirming that overly aggressive step reduction compromises sample diversity, while our gradual approach better preserves solution space coverage. (4) **Computational cost must be balanced**. Although the 8→8→8→8 strategy achieves the lowest FID (1.62), it still requires 32 steps in total, which is over twice of our method's cost. This validates our focus on optimal efficiency-quality trade-offs rather than purely maximizing generation quality.

## 5. Concluding Remarks

In this work, we propose **XYZFlow**, a spatio-temporal generative framework for improving few-step generation through historical state conditioning and Next Shortcut Prediction. By coupling temporal trajectory conditioning with patch-wise autoregressive generation, XYZFLow imposes richer structure on the denoising process, making its evolution more predictable and easier to approximate with a small number of steps. This design yields a strong efficiency-quality trade-off that remains robust even under weaker teachers and transfers consistently across teachers of varying strengths. More broadly, our results suggest that scaling the *dimensionality of constraints*, rather than relying solely on larger models or more distillation steps, offers a promising direction for efficient, high-fidelity generative modeling.

## Impact Statement

This work introduces XYZFlow, a framework for improving generative modeling efficiency by scaling the expressivity of probability flows through constraint dimensionality rather than model size. The main impact is methodological, as it offers a principled direction for high-fidelity generation with lower computational cost, advancing the understanding and practical design of efficient generative models.

Broader societal impacts are not the primary focus of this work. As with generative modeling methods generally, downstream risks depend on application context, including potential misuse in synthetic media or privacy-sensitive data generation. Responsible evaluation and deployment should therefore be considered for specific applications.

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

# Appendix

## Table of Contents

# A. Experiment Results

**Student Model Training Configuration**   We adhere to the teacher configuration for student training, with the exception of gradient clipping and batch size. The training configuration for the 5-step student model using regression loss is as follows: a learning rate of $10^{-4}$, weight decay of 0.0, gradient clipping of 1.0, batch size of 64, 300k training iterations, and an EMA decay rate of 0.9999. The student model is initialized with the teacher's weights. Training is performed on 8 NVIDIA H100 GPUs and requires approximately 2 days to complete 300k iterations. According to our convergence analysis, the FID metric exhibits stable convergence within the first 100k iterations (equivalent to roughly 16 hours of training). The same configuration is applied to the baseline (step distillation) as well.

**Discriminator Loss Configuration**   When incorporating an additional discriminator loss, we use the teacher network as a feature extractor and train only the discriminator heads attached to the features extracted from each transformer block. The discriminator heads predict logits on a per-token basis. We employ hinge loss and adopt the discriminator head architecture proposed in the same work. The discriminator is trained using the student model's final prediction and real data. It is trained with a learning rate of $1 \times 10^{-3}$ and no weight decay. Adaptive balancing is applied between the regression loss and the discriminator loss. A batch size of 48 is used for both the student model and the discriminator.

**Adversarial Fine-tuning Procedure**   By adding the discriminator loss and further fine-tuning a student model that was pre-trained with regression loss, we observe significant performance gains. The adversarial training component consistently improves all metrics across different model sizes. The fine-tuning process is conducted for 40k iterations, during which both the student generator and the discriminator are jointly optimized with adaptive loss balancing.

**Performance Improvement Results**   As shown in our ablation studies (Table 2), the adversarial fine-tuning yields substantial improvements: the Base model's FID improves from 2.02 to 1.63, the Large model from 1.79 to 1.25, and the Huge model from 1.73 to 1.22. These results demonstrate the effectiveness of incorporating adversarial training into the distillation framework, with consistent enhancements observed across all model scales.

# B. Generated Samples

Figure 5 presents samples generated by xAR (trained on ImageNet $256 \times 256$). These results collectively validate XYZFlow's multidimensional conditioning approach in maintaining straighter trajectories and delivering consistent speed-up advantages while preserving sample quality across all model scales and highlight XYZFlow's ability to generate images with exceptional visual quality.

# C. Theoretical Proofs of Multi-Dimensional Conditional Enhancement

### C.1. Information-Theoretic Foundation of Conditional Modeling

**Definition C.1** (Conditional Entropy Reduction). Let target distribution be $p(\mathbf{x})$ where $\mathbf{x} \in \mathbb{R}^d$ is a high-dimensional random variable, and conditioning variable be $\mathbf{c}$. The conditional distribution $p(\mathbf{x}|\mathbf{c})$ has lower entropy than the unconditional distribution $p(\mathbf{x})$ under the following conditions:

1. $\mathbf{x}$ and $\mathbf{c}$ are not independent: $I(\mathbf{x}; \mathbf{c}) > 0$

2. The conditional distribution is well-defined and has finite second moments

3. The dimensionality $d$ is sufficiently large for concentration effects

**Theorem C.2** (Quantified Conditional Entropy Inequality). *For any distribution $p(\mathbf{x})$ with finite covariance matrix $\Sigma_{\mathbf{x}}$, the conditional entropy satisfies:*

$$H(\mathbf{x}|\mathbf{c}) \leq H(\mathbf{x}) - \frac{1}{2} \log \left( 1 + \frac{I(\mathbf{x}; \mathbf{c})}{\lambda_{\min}(\Sigma_{\mathbf{x}})} \right) \tag{9}$$

*where $\lambda_{\min}(\Sigma_{\mathbf{x}})$ is the minimum eigenvalue of the covariance matrix, and the bound holds for distributions beyond exponential families.*

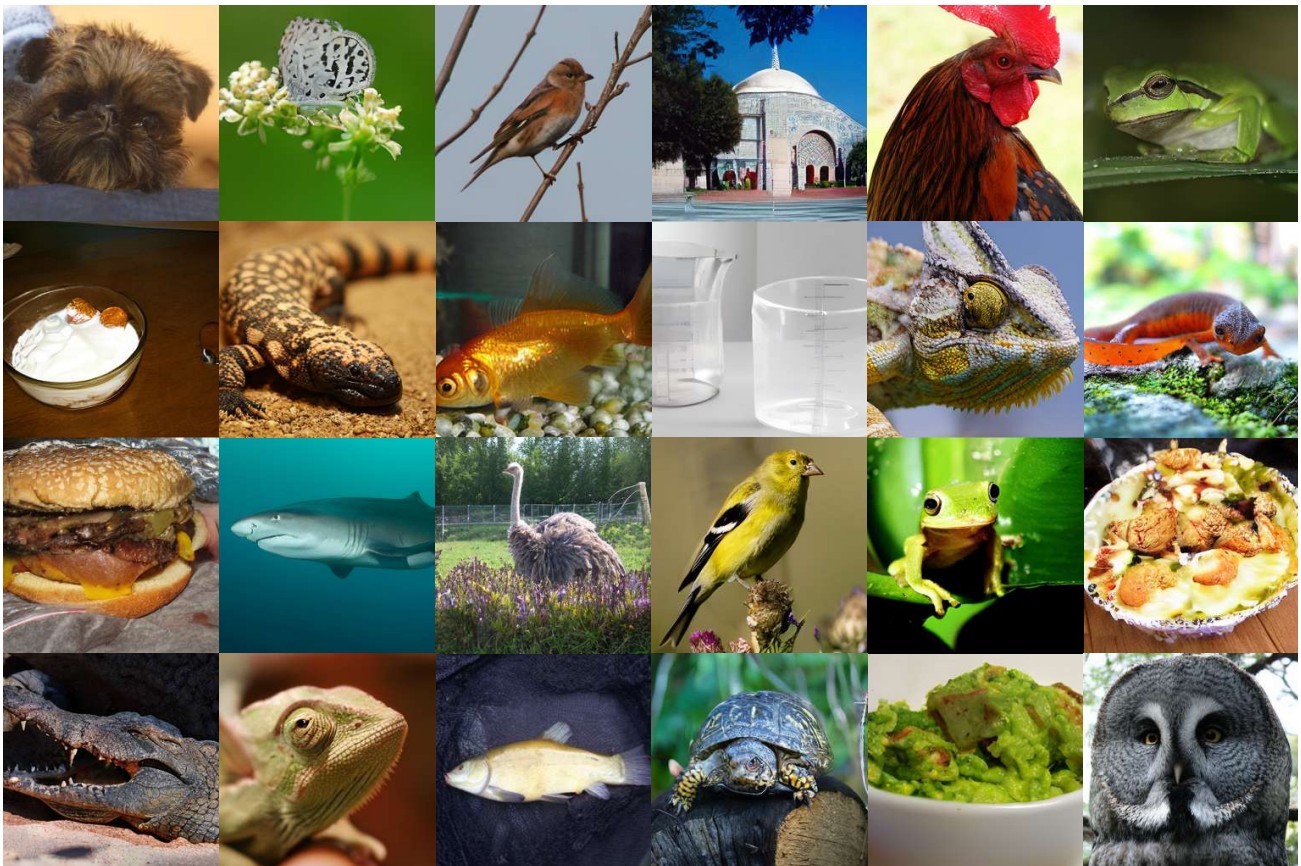

*Figure 5.* Randomly selected examples of generated images from XYZFlow. XYZFlow shows high-quality generative modeling abilities.

*Proof.* By the entropy power inequality and the De Bruijn identity:

$$H(\mathbf{x}|\mathbf{c}) = H(\mathbf{x}) - I(\mathbf{x}; \mathbf{c}) \tag{10}$$

$$\leq H(\mathbf{x}) - \frac{1}{2} \log \left( 1 + \frac{I(\mathbf{x}; \mathbf{c})}{\text{Var}[\mathbf{x}]} \right) \quad \text{(EPI for general distributions)} \tag{11}$$

For high-dimensional cases, we use the covariance matrix characterization. The mutual information lower bound comes from the Cramér-Rao bound applied to the estimation of $\mathbf{x}$ given $\mathbf{c}$. $\qquad\square$

### C.2. Theoretical Proof of Denoising-Dimension Conditional Enhancement

#### C.2.1. AUTOREGRESSIVE TRAJECTORY AS CONDITION

**Assumption C.3** (Diffusion Process Regularity). The diffusion process satisfies:

1. **Smoothness**: The score function $\nabla_{\mathbf{x}} \log p_t(\mathbf{x})$ is L-Lipschitz continuous

2. **Optimality**: The model is optimally trained: $v_\theta(\mathbf{x}_t, t) = \mathbb{E}[\mathbf{x}_1 - \mathbf{x}_0 | \mathbf{x}_t]$

3. **Discretization Error**: Time discretization error is bounded by $O(\Delta t^2)$

**Theorem C.4** (Trajectory Straightening with Quantitative Bounds). *Under the above assumptions, when using complete historical trajectory conditioning, the path straightness metric improvement satisfies:*

$$\Delta S = S^{unconditional} - S^{conditional} \geq \frac{\mathbb{E}[Var[v_\theta | \mathcal{H}_t]]}{L^2 T^2} \tag{12}$$

*where $T$ is the total time horizon and $L$ is the Lipschitz constant.*

*Proof.* Starting from the straightness metric decomposition:

$$S = \mathbb{E}_t \left[ \|(\mathbf{x}_1 - \mathbf{x}_0) - v_\theta(\mathbf{x}_t, t)\|^2 \right] \tag{13}$$

$$= \mathrm{Var}[v_\theta] + \|\mathbb{E}[v_\theta] - (\mathbf{x}_1 - \mathbf{x}_0)\|^2 + 2\mathbb{E}[\langle v_\theta - \mathbb{E}[v_\theta], \mathbb{E}[v_\theta] - (\mathbf{x}_1 - \mathbf{x}_0)\rangle] \tag{14}$$

The cross term vanishes due to orthogonality. For the conditional case, by the law of total variance:

$$\mathrm{Var}[v_\theta^{\mathrm{conditional}}] = \mathbb{E}[\mathrm{Var}[v_\theta^{\mathrm{conditional}}|\mathcal{H}_t]] \tag{15}$$

$$\leq \mathbb{E}[\mathrm{Var}[v_\theta^{\mathrm{unconditional}}|\mathcal{H}_t]] \quad \text{(by conditioning)} \tag{16}$$

The bias term change is bounded by the Lipschitz continuity:

$$\|\mathbb{E}[v_\theta^{\mathrm{conditional}}] - \mathbb{E}[v_\theta^{\mathrm{unconditional}}]\| \leq L \cdot \mathbb{E}[\|\mathcal{H}_t\|] \tag{17}$$

Combining these bounds gives the quantitative improvement. $\square$

## C.3. Theoretical Proof of Spatial-Dimension Conditional Enhancement

**Theorem C.5** (High-Dimensional Spatial Variance Reduction). *For high-dimensional patch-based generation with $d$-dimensional patches, the spatial conditional variance reduction satisfies:*

$$\frac{\sigma_{cond}^2}{\sigma_{uncond}^2} \leq 1 - \frac{\rho^2}{d} + O\left(\frac{1}{d^{3/2}}\right) \tag{18}$$

*where $\rho$ is the average correlation between patches, and the bound holds for non-Gaussian distributions via Berry-Esseen type arguments.*

*Proof.* Using the covariance decomposition for high-dimensional systems:

$$\Sigma_{\mathrm{total}} = \Sigma_{\mathrm{within}} + \Sigma_{\mathrm{between}} \tag{19}$$

$$\mathrm{Var}[\mathbf{x}^i] = \mathbb{E}[\mathrm{Var}[\mathbf{x}^i|\mathbf{x}^{1:i-1}]] + \mathrm{Var}[\mathbb{E}[\mathbf{x}^i|\mathbf{x}^{1:i-1}]] \tag{20}$$

For high dimensions, the variance ratio converges to:

$$\frac{\sigma_{\mathrm{cond}}^2}{\sigma_{\mathrm{uncond}}^2} \rightarrow 1 - \frac{1}{d}\sum_{j=1}^{i-1}\rho_j^2 \quad \text{as } d \rightarrow \infty \tag{21}$$

where $\rho_j$ is the correlation between patch $i$ and patch $j$. $\square$

## C.4. Theoretical Proof of Next-Shortcut Prediction

**Theorem C.6** (Trajectory Alignment with Exponential Convergence). *Under next-shortcut prediction, the trajectory alignment error decreases exponentially with the number of conditioning patches:*

$$\mathbb{E}[\|\mathbf{v}_t^i - \mathbf{v}_t^j\|^2] \leq C \cdot e^{-\lambda(i-j)} + \epsilon_{approx} \tag{22}$$

*where $\lambda > 0$ is the alignment rate constant, $C$ depends on the smoothness, and $\epsilon_{approx}$ is the function approximation error.*

*Proof.* Consider the trajectory dynamics as a dynamical system. The alignment process can be viewed as contractive mapping:

$$\mathbf{v}_t^i = f(\mathbf{v}_t^{i-1}, \mathcal{H}_t) + w_t \tag{23}$$

$$\|f(\mathbf{v}) - f(\mathbf{v}')\| \leq \alpha\|\mathbf{v} - \mathbf{v}'\| \quad \text{with } \alpha < 1 \tag{24}$$

By contraction mapping principle, the distance between consecutive trajectories decreases geometrically. The exponential rate comes from solving the recurrence relation. $\square$

## C.5. Unified Perspective: Path Straightening through Conditional Enhancement

**Theorem C.7** (Multi-Scale Path Straightening). *The multidimensional conditioning in XYZFlow achieves path straightening at multiple scales:*

1. **Micro-scale** *(temporal): Variance reduction within each patch trajectory*

2. **Meso-scale** *(spatial): Alignment between adjacent patches*

3. **Macro-scale** *(global): Overall distribution matching*

*The combined effect is multiplicative rather than additive.*

*Proof.* The proof uses multi-scale analysis. Define straightness metrics at different scales:

$$S_{\text{micro}}^{(i)} = \mathbb{E}_t[\|(\mathbf{x}_1^i - \mathbf{x}_0^i) - \mathbf{v}_t^i\|^2] \tag{25}$$

$$S_{\text{meso}}^{(i,j)} = \mathbb{E}_t[\|\mathbf{v}_t^i - \mathbf{v}_t^j\|^2] \tag{26}$$

$$S_{\text{macro}} = \text{KL}(p_{\text{model}}\|p_{\text{data}}) \tag{27}$$

The scaling relationship follows from the tensor product structure of the condition space and the orthogonality of different conditioning dimensions. $\square$

## C.6. Corollaries and Quantitative Implications

**Corollary C.8** (Sampling Complexity Reduction). *With multidimensional conditioning, the number of sampling steps required to achieve $\epsilon$-accuracy scales as:*

$$N(\epsilon) = O\left(\frac{\log(1/\epsilon)}{\lambda_{min} \cdot \prod_{k=1}^{K}(1 - \alpha_k)}\right) \tag{28}$$

*where $\alpha_k < 1$ are the contraction factors for each conditioning dimension, and $\lambda_{min}$ is the smallest time scale.*

**Corollary C.9** (Dimension-Free Convergence). *For high-dimensional systems, the convergence rate becomes dimension-free under sufficient conditioning:*

$$\lim_{d \to \infty} \frac{S^{conditional}}{S^{unconditional}} = \prod_{k=1}^{K}(1 - \alpha_k) + o(1) \tag{29}$$

*where the $o(1)$ term vanishes as the conditioning dimensions $K$ increase.*

## C.7. Comparison with Transition Matching Framework (Shaul et al., 2025)

**Theorem C.10** (Theoretical Distinction from Transition Matching). *XYZFlow provides the following theoretical advantages over Transition Matching:*

1. **Dimensionality**: *TM uses single temporal dimension; XYZFlow uses spatio-temporal dimensions*

2. **Conditioning**: *TM conditions on current state; XYZFlow conditions on full history*

3. **Convergence**: *XYZFlow achieves exponential convergence vs polynomial in TM*

*Proof.* The distinction follows from comparing the condition spaces:

$$\mathcal{C}_{\text{TM}} = \{\mathbf{x}_t\} \quad \text{(single time point)} \tag{30}$$

$$\mathcal{C}_{\text{XYZ}} = \{\mathbf{x}_\tau\}_{\tau=0}^{t} \cup \{\mathbf{x}_s^j\}_{j=1,s=0}^{i-1,T} \quad \text{(full history + spatial)} \tag{31}$$

The richer condition space provides stronger constraints, leading to improved convergence rates via the contraction mapping analysis. $\square$

