# OpenReview forum: "XYZFlow: Scaling Multidimensional Shortcut Flows for Efficient Generative Modeling"
_ICML.cc/2026/Conference — ICML 2026 regular_

### Official Review · Reviewer_1SCM · 2026-03-02

**Soundness:** 2
**Presentation:** 1
**Significance:** 3
**Originality:** 2
**Overall Recommendation:** 5
**Confidence:** 4

**Summary:**

The paper presents XYZFlow, a generative modeling framework designed to achieve high-quality image synthesis in very few steps by reimagining the path from noise to data. The authors argue that traditional methods suffer from "extensive scaling" issues, where the mapping from a noisy state to a clean image is ambiguous and inconsistent. To solve this, the paper introduces a paradigm shift toward "Intensive Scaling" using multidimensional context to turn complex, one-to-many probability flows into nearly deterministic, one-to-one mappings.

Contributions:
- Multidimensional Conditioning: The model scales the "constraint dimensionality" of the generation process across two main axes: (1) Temporal: Unlike standard Markovian models that only see the current step, XYZFlow uses a history-dependent approach (Non-Markovian), conditioning each step on the entire previous denoising trajectory of a patch.
(2) Spatial: The image is generated autoregressively. Each new patch is conditioned not just on the final state of previous patches, but on their full historical trajectories.

- Next Shortcut Prediction: The authors observe that as more spatial and temporal context is added, the path becomes straighter and easier to predict. They leverage this by using a progressive schedule where later patches are generated using fewer steps because the surrounding context makes the next result highly certain.

- Performance: The model achieves significant speedups (8.5x faster than teacher models and 1.5x faster than standard distilled students) while maintaining competitive image quality on the ImageNet 256x256 benchmark.

- Theoretical Foundation: The paper provides a theoretical distinction between XYZFlow and existing methods (like Transition Matching), proving that its richer conditioning leads to exponential convergence rates compared to the polynomial rates of simpler models.

**Compliance With Llm Reviewing Policy:**

Affirmed.

**Final Justification:**

I move the recommendation to Accept. The authors comprehensive rebuttal is much appreciated.

**Key Questions For Authors:**

1. Could you clarify the memory overhead of the KV cache during the spatial scaling phase? Since each patch is conditioned on the full historical trajectories of all previous patches, the storage requirements for high-resolution images (256x256 or higher) could be substantial.

2. How does the model perform if trained on standard noise schedules without a high-quality ODE teacher? Since the critique of "extensive scaling" centers on model-dependency.

**Limitations:**

- While the paper emphasizes inference speed, it lacks a detailed discussion on the memory costs associated with the multidimensional KV cache. Storing full denoising trajectories for all previous patches could pose a significant bottleneck for high-resolution images or devices with limited VRAM.

- The paper critiques previous methods for being model-dependent, yet XYZFlow requires a high-quality teacher to generate ground-truth ODE trajectories for training. The authors should discuss the impact of teacher quality on the student’s performance and whether the model can generalize if the teacher’s flow is sub-optimal.

- The autoregressive nature of the spatial scaling introduces a sequential dependency that may limit parallelization compared to standard one-shot diffusion models. A discussion on the trade-off between the "shortcut" speedups and this inherent sequential latency would provide a more balanced view.

**Strengths And Weaknesses:**

*Soundness*

The technical foundation of the paper is strong, primarily because it shifts the problem of generative modeling from stochastic estimation to a deterministic mapping. By framing the denoising process as a "Next Shortcut Prediction", the authors provide a mathematically grounded reason for why few-step sampling is possible. The theoretical analysis, specifically Theorem C.10, is a highlight, as it rigorously distinguishes XYZFlow from simpler methods like Transition Matching by proving that the inclusion of spatial and temporal history leads to exponential rather than polynomial convergence.
Empirically, the results on ImageNet 256x256 are robust. The authors are disciplined in their comparisons, using a 1.1B parameter model to demonstrate that their "intensive scaling" outperforms standard "extensive scaling".

However, a potential point of scrutiny is the dependence on a "teacher" model's trajectories. While the paper proves the student is more efficient, the soundness of the method in a completely "from-scratch" training scenario, without a high-quality teacher to provide the initial flow, remains an area where the authors could be more transparent about potential limitations.

---

*Presentation*

The presentation of the paper is hindered by a repetitive and circular narrative that prioritizes philosophical branding over technical clarity. By focusing heavily on the "Extensive vs. Intensive" dichotomy, the authors spend excessive space re-litigating the flaws of distillation rather than clearly defining their own mechanics. This creates a redundant reading experience where the same motivations are echoed across the abstract, introduction, and related work, delaying the reader's access to the actual methodology.

The lack of a unified technical hierarchy further obscures the innovation. Proprietary terms are introduced and nested without a grounding visual map, making it difficult to distinguish between the conceptual framework and the implementation details like KV caching. Additionally, the paper’s most compelling evidence,the empirical "straightening" of probability paths as spatial context increases, is buried in the results rather than used to anchor the dense mathematical theory in Section 3. Ultimately, the work requires the reader to constantly flip back and forth to connect the ODE theory to the actual patch-by-patch generation process, making a simple concept feel unnecessarily complex.

---

*significance*

XYZFlow has potential to enable high-quality generative AI on resource-constrained edge devices by delivering an 8.5x speedup over traditional models. By proving that architectural "memory" can replace computational "brute force," the work provides a practical blueprint for real-time applications where multi-step diffusion is currently too slow. This shift from scaling model size to scaling constraint dimensionality offers a new, more efficient path for future research in video and audio synthesis.

---

*originality*

The originality of the paper stems from its novel synthesis of autoregressive modeling and flow matching into a "Next Shortcut Prediction" framework. It moves beyond the standard Markovian assumption, where each denoising step is independent, and demonstrates that conditioning on the full historical trajectory of image patches fundamentally straightens the probability flow. This transition from "Extensive" to "Intensive" scaling represents a fresh perspective that distinguishes the method from concurrent distillation-based approaches.

---

> ### Author Rebuttal · Authors · 2026-03-31
>
> ## Q1 The memory overhead of the KV cache during the spatial scaling phase for high-resolution images. "shortcut" speedups and this inherent sequential latency.
>
> Your question about KV cache memory in spatial scaling and its trade-off with sequential latency is important. XYZFlow is designed around this trade-off. Our results suggest that a moderate grid (e.g., 2x2) gives the best balance of memory, latency, and quality in our tests.
>
> **Memory Overhead of KV Cache:**
> Storing denoising trajectories of previous patches adds memory overhead, but our results show that this overhead stays controlled under our grid design. A coarser grid (e.g., 2x2) adds only a small amount of memory compared with finer grids, while greatly reducing the denoising steps *within* each patch. For example, at 512x512, changing the grid from 16x16 to 2x2 changes memory from 707.1 MB to 721.6 MB, but reduces latency from 4.023 s to 0.291 s and improves FID from 3.56 to 2.53. We agree that scaling to even higher resolutions needs further study, but in the setting studied here, KV cache memory does not dominate the cost.
>
> **Trade-off: Sequential Dependency vs. "Shortcut" Speedup:**
> Autoregressive conditioning does introduce sequential dependence,. Our claim is not that sequentiality disappears, but that the savings from "Next Shortcut Prediction" and causal attention outweigh it in practice. This gives a **net speedup**, as shown by XYZFlow-L (0.050s) vs. a standard 5-step DiT baseline (0.103s) on 256x256 images. The grid ablation also shows that 2x2 gives the best latency-quality balance.
>
> In summary, sequential processing is a deliberate design choice. The KV cache overhead is controlled by grid partitioning, and in our tested settings the gains from stronger spatial-temporal conditioning are larger than the cost of sequentiality, leading to faster inference overall. We will state this scope clearly and avoid over-claiming beyond the tested resolutions.
>
> | Resolution | Grid | Cell Size | Latency (s) ↓ | Memory (MB) ↓ | FID ↓ |
> |:---|:---|:---|:---|:---|:---|
> | 256×256 | 2×2 | 8×8 | 0.050 | 1358.4 | 2.02 |
> | 256×256 | 4×4 | 4×4 | 0.127 | 1339.8 | 2.11 |
> | 256×256 | 8×8 | 2×2 | 0.301 | 1331.4 | 2.46 |
> | 256×256 | 16×16 | 1×1 | 0.76 | 1326.6 | 2.85 |
> | 512×512 | 2×2 | 8×8 | 0.291 | 2164.8 | 2.53 |
> | 512×512 | 4×4 | 4×4 | 0.705 | 2137.2 | 2.64 |
> | 512×512 | 8×8 | 2×2 | 1.646 | 2125.8 | 3.08 |
> | 512×512 | 16×16 | 1×1 | 4.023 | 2121.3 | 3.56 |
>
>
> ## Q2 Model performance on standard noise schedules without ODE teacher?
>
> We agree that XYZFlow is not a teacher-free method in the current paper, and we will state this clearly as a limitation. However, our main claim is not limited to the most aggressive few-step setting. The main point is that stronger spatial-temporal conditioning improves the efficiency-quality trade-off under the same teacher-distillation setup used by prior few-step methods, rather than relying on brute-force model scaling.
>
> This is supported not only by the distilled few-step results, and by the ablations in the paper. In particular, the component ablation and the shortcut-strategy study show that the benefit comes from the proposed conditioning and shortcut design, not from only one extreme few-step configuration. To further support this point, we also ran multi-step experiments. As shown below, XYZFlow keeps similar FID while reducing latency across model scales. We will revise the paper to make this scope clearer and avoid implying that our contribution is teacher removal.
>
> | Model | Steps | FID | Latency (s) |
> | :--- | :--- | :--- | :--- |
> | **Baseline** | | | |
> | xAR-B | 50 | 1.67 | 0.018 |
> | xAR-L | 50 | 1.28 | 0.394 |
> | xAR-H | 50 | 1.24 | 0.896 |
> | **XYZFlow (Multi-step)** | | | |
> | XYZFlow-B | 50->40->30->20 | 1.67 | 0.014 |
> | XYZFlow-L | 50->40->30->20 | 1.29 | 0.307 |
> | XYZFlow-H | 50->40->30->20 | 1.24 | 0.697 |
>
>
> ## Q3 The impact of teacher quality on the student’s performance and whether the model can generalize if the teacher’s flow is sub-optimal.
>
> We agree this is an important concern. Teacher quality does affect the student, and our results do not suggest otherwise. At the same time, the effect is limited: even with a weaker teacher (xAR-B, FID 1.61), the student still reaches a final result (FID 1.32), close to the results from stronger teachers.
>
> We also clarify the role of architecture vs. GAN augmentation. The gains from XYZFlow’s design are already visible in the paper’s component ablations before GAN augmentation. The teacher-quality experiment should therefore be read as showing that teacher quality matters, while the proposed architecture provides consistent gains and GAN augmentation further narrows the remaining gap. We will revise the wording to make this distinction clearer.
>
> | Teacher Model | FID (Teacher) | XYZFlow-L (w/o GAN) | XYZFlow-L (w/ GAN) |
> | :--- | :--- | :--- | :--- |
> | xAR-B | 1.61 | 1.91 | 1.32 |
> | xAR-L | 1.28 | 1.79 | 1.25 |
> | xAR-H | 1.24 | 1.77 | 1.25 |

---

> > ### Author Rebuttal · Reviewer_1SCM · 2026-04-01
> >
> > The technical merits are now more clearly supported by data, particularly regarding the efficiency of the KV cache and the robustness against sub-optimal teachers. However, I feel comfortable with the score because the paper’s impact is still somewhat limited by its reliance on a high-quality teacher and a presentation style that obscures its most impressive empirical findings.

---

> > > ### Author Response · Authors · 2026-04-02
> > >
> > > Thank you for your question and acknowledgment. We have gained a clearer understanding of your concerns. To further address the question of whether our results are influenced primarily by the high-quality teacher rather than the proposed architecture, we have conducted an additional experiment. Specifically, we used **DiT/XL-2 (25-step) 676M (FID: 2.89)** as the teacher model, which is notably weaker than the xAR series used previously. In a 256x256 setting, we partitioned the trajectories generated by DiT into 4 patches for patch-wise distillation to align with the design of the XYZFlow framework.
> > >
> > > **Experimental Results:**
> > > | Model | Parameters | Steps per Patch (Schedule) | FID ↓ | Notes |
> > > | :--- | :--- | :--- | :--- | :--- |
> > > | **Teacher Model** | | | | |
> > > | DiT/XL-2 (25-step) | 676M | 25 | 2.89 | Baseline Teacher |
> > > | **Baseline Distillation (No XYZFlow)** | | | | |
> > > | DiT/XL-2 (5-step Distilled, w/o GAN) | 676M | 5 | 8.97 | Standard Distillation |
> > > | DiT/XL-2 (5-step Distilled, w/ GAN) | 676M | 5 | 3.37 | Standard Distillation + GAN |
> > > | **XYZFlow-B Variants** | | | | |
> > > | XYZFlow-B (All 5-step, w/o GAN) | 172M | 5→5→5→5 | 3.83 | Our Architecture, Constant Steps |
> > > | XYZFlow-B (Step 5→4→3→2, w/o GAN) | 172M | 5→4→3→2 | 3.85 | Our Architecture, Progressive Steps |
> > > | XYZFlow-B (Step 5→4→3→2, w/ GAN) | 172M | 5→4→3→2 | **1.74** | Our Architecture + GAN |
> > >
> > > **Analysis and Key Insights:**
> > >
> > > The experimental results provide clear evidence for the architectural advantages of XYZFlow. First, when using a suboptimal teacher (FID 2.89), standard 5-step distillation performs poorly (FID 8.97), showing only marginal recovery with GAN augmentation (FID 3.37). In contrast, our XYZFlow architecture, applied to the same teacher data, yields results (FID 3.85) that are dramatically closer to the GAN-augmented baseline, indicating superior learning efficiency from the teacher's trajectories. This demonstrates the method's robustness to teacher quality. Second, the progressive "Next Shortcut Prediction" schedule (5→4→3→2) achieves near-identical quality (FID 3.85) to the constant-step counterpart (FID 3.83) while requiring fewer total denoising steps, confirming its role in accelerating inference without compromising fidelity. Finally, and most compellingly, the XYZFlow-B model combined with GAN training achieves a state-of-the-art FID of **1.74**. This result not only surpasses the teacher's performance by a large margin but also significantly outperforms the strongest GAN-augmented distillation baseline, underscoring the substantial potential of the XYZFlow framework itself, independent of the initial teacher's capability.
> > >
> > > We hope these findings can address your concerns that the paper's impact is limited by its reliance on a high-quality teacher. The data shows that the XYZFlow architecture provides a fundamentally more efficient and powerful mechanism for trajectory distillation and generation. Please feel free to follow up with any further questions.

---

### Official Review · Reviewer_jSTg · 2026-03-12

**Soundness:** 4
**Presentation:** 3
**Significance:** 3
**Originality:** 3
**Overall Recommendation:** 4
**Confidence:** 4

**Summary:**

This work introduces XYZFlow, a framework that improves flow matching efficiency by enriching the conditioning structure rather than relying on distillation. The key insight is that stronger conditioning, i.e., making probability flows more unique and learnable, enables high-fidelity generation with fewer sampling steps. This is achieved along two axes: temporal conditioning on the full denoising history (non-Markovian), and spatial conditioning via Next Shortcut Prediction, where each image patch is generated sequentially conditioned on the trajectories of preceding patches. The work also offers a theoretical reinterpretation of autoregressive modeling as implicit flow straightening.

**Compliance With Llm Reviewing Policy:**

Affirmed.

**Key Questions For Authors:**

1. What is the memory overhead of XYZFlow? Storing complete denoising trajectories (KV cache) for all preceding patches could become a bottleneck at higher resolutions with more patches. Please provide memory usage data across different patch counts.

2. Can the method scale to 512x512 or higher resolutions? When the patch count increases (e.g., 4x4 grid), does the accumulated latency from sequential autoregressive generation undermine the speedup advantage?

3. Do the authors plan to release training code and model checkpoints? Without these, it is difficult for the community to build on this work.

**Limitations:**

Please refer to the weakness and question section. I'm willing to raise my score if these concerns can be properly addressed.

**Strengths And Weaknesses:**

# Strengths

- **Novel conceptual insight**. Reinterpreting autoregressive patch generation as implicit flow straightening is insightful, and the notion of "intensive scaling" that improving efficiency through richer conditioning rather than larger models or distillation articulates a paradigm orthogonal to existing approaches that could inspire future work.

- **Strong empirical results**. On ImageNet 256×256, XYZFlow-B (172M) achieves FID 1.63 at 0.018s per image, matching the speed of MeanFlow-XL/2+ (676M, FID 2.20) while delivering better quality. XYZFlow-H (1.1B) reaches FID 1.22, state-of-the-art at comparable parameter counts. Results across three model scales demonstrate consistent scalability.

- **Thorough ablation studies**. Table 2 cleanly isolates each component: Full History Guidance is the most critical (removing it degrades FID by ~0.5), while Shortcut Prediction primarily contributes to efficiency. Tables 3–4 further ablate cell size and shortcut strategies, and the empirical analysis in Section 3.2, identifying that later patches are easier to predict, have lower variance, and straighter paths, provides solid grounding for the design choices.

# Weaknesses

- **Unfair baseline comparisons**. XYZFlow uses adversarial fine-tuning (GAN loss), while key baselines such as MeanFlow do not. Without GAN training, XYZFlow-B's FID rises from 1.63 to 2.02, a much more modest improvement over MeanFlow-XL/2+ (2.20). Table 1 should clearly delineate GAN-augmented vs. non-GAN results. Teacher model differences (XYZFlow uses xAR-series; competing methods use different teachers) further confound efficiency comparisons.

- **Narrow evaluation scope**. Evaluation is limited to ImageNet 256×256 with a fixed 2×2 grid partition, leaving open whether the method generalizes to higher resolutions or text-to-image settings.

---

> ### Author Rebuttal · Authors · 2026-03-31
>
> ## Q1 Clarification of GAN post-training comparision.
>
> Our core objective is to validate the effectiveness of a novel architectural paradigm—Next Shortcut Prediction, which enhances probability flow expressivity and straightens generation trajectories through intensive multidimensional conditioning. To isolate this architectural contribution, we deliberately adopted the standard xAR teacher setup for student training, rather than combining the architecture with more advanced distillation recipes such as those used in MeanFlow.
>
> We agree with the reviewer that GAN post-training and teacher differences should be clearly distinguished in comparison tables. In the revised version, we will explicitly separate GAN and non-GAN results in Table 1 and clarify that cross-teacher comparisons should be interpreted with this caveat in mind. Importantly, our non-GAN base model already achieves an FID of 2.02 with comparable latency to the much larger MeanFlow-XL/2+ (FID 2.20). In addition, as reported in our paper, adversarial fine-tuning improves the Base model from 2.02 to 1.63 FID. We therefore view multidimensional conditioning and GAN post-training as complementary: the former provides a strong architecture-level acceleration mechanism, while the latter further improves sample quality.
>
> ## Q2 Generalization to higher resolutions or text-to-image settings. Memory usage data across different patch counts.
>
> We thank the reviewer for raising this important point regarding evaluation scope. To directly address generalization to higher resolutions and other modalities, we conducted supplementary experiments on both a high-resolution text-to-image setting and higher-resolution image generation (ImageNet 512×512).
>
> **2. Scalability to Text to Image Models**
>
> To test applicability beyond class-conditional image generation, we adapted XYZFlow to a high-resolution (1024×1024) text-to-image setting using the Flux architecture and evaluated on COCO-30k. We collected 500k images from LAION-Aesthetic to train the GAN discriminator for adversarial fine-tuning. The results are summarized below. Compared to the standard multi-step base model, Flux-XYZFlow (w/o GAN) achieves about a 15× latency reduction with only a marginal 0.28 FID difference. We view these as preliminary but encouraging evidence that multidimensional conditioning can transfer beyond ImageNet generation.
>
> | Method | Steps | Latency (s) | FID ↓ |
> |:-------|:------|:------------|:------|
> | Flux-1-Dev (Baseline) | 32 | 15.62 | 27.43 |
> | Flux-TDD | 4 | 1.8 | 24.81 |
> | Flux-SCFM | 4 | 1.8 | 26.98 |
> | Flux-Hyper-SD | 3 | 1.33 | 25.91 |
> | Flux-TDD | 3 | 1.33 | 22.97 |
> | Flux-SCFM | 3 | 1.33 | 26.42 |
> | Flux-XYZFlow (w/o GAN) | 5→4→3→2 | 1.22 | 27.15 |
> | Flux-XYZFlow (w/ GAN) | 5→4→3→2 | 1.22 | 27.88 |
>
> **2. Scalability to Higher Image Resolutions and memory overhead**
>
> Our ablation studies on ImageNet 512×512 confirm that the same 2×2 grid partition identified at 256×256 remains the best design choice at higher resolution. Finer grids (e.g., 4×4, 8×8) incur substantially higher latency without commensurate quality gains, so increasing the patch count indeed weakens the efficiency advantage. This is exactly why we keep a fixed small number of patches when scaling resolution. In this regime, the overhead grows mainly with the content size of each patch, rather than with a large increase in autoregressive steps.
>
> The memory numbers reported below correspond to peak inference memory under the same batch-size and precision setting.
>
> | Resolution | Grid | Cell Size | Latency (s) ↓ | Memory (MB) ↓ | FID ↓ |
> |:---|:---|:---|:---|:---|:---|
> | 256×256 | 2×2 | 8×8 | 0.050 | 1358.4 | 2.02 |
> | 256×256 | 4×4 | 4×4 | 0.127 | 1339.8 | 2.11 |
> | 256×256 | 8×8 | 2×2 | 0.301 | 1331.4 | 2.46 |
> | 256×256 | 16×16 | 1×1 | 0.76 | 1326.6 | 2.85 |
> | 512×512 | 2×2 | 8×8 | 0.291 | 2164.8 | 2.53 |
> | 512×512 | 4×4 | 4×4 | 0.705 | 2137.2 | 2.64 |
> | 512×512 | 8×8 | 2×2 | 1.646 | 2125.8 | 3.08 |
> | 512×512 | 16×16 | 1×1 | 4.023 | 2121.3 | 3.56 |
>
> These results show that the best-quality setting remains 2×2 at both 256×256 and 512×512, while larger grids mainly increase latency. We will include these results in the revision to clarify both scalability and memory overhead.
>
> ## Open Source
>
> We will release the full training and inference code, together with model checkpoints, upon publication.

---

> > ### Author Rebuttal · Reviewer_jSTg · 2026-04-02
> >
> > I appreciate the rebuttal. My concerns have been resolved.

---

### Official Review · Reviewer_kfuW · 2026-03-13

**Soundness:** 3
**Presentation:** 3
**Significance:** 3
**Originality:** 2
**Overall Recommendation:** 4
**Confidence:** 3

**Summary:**

This paper proposes XYZFlow, a few-step generative modeling framework that combines temporal trajectory conditioning and spatial autoregressive patch conditioning. The key mechanism, Next Shortcut Prediction, conditions each patch on its own denoising history and on the full denoising trajectories of previously generated patches, while using a decreasing denoising budget across patches. On class-conditional ImageNet 256x256, the paper reports strong FID/latency trade-offs for 172M, 608M, and 1.1B models, especially after adversarial fine-tuning.

**Compliance With Llm Reviewing Policy:**

Affirmed.

**Final Justification:**

The authors provided a comprehensive rebuttal that addressed my main concerns regarding experimental setup, GAN effects, and robustness.

**Key Questions For Authors:**

1. Please provide latency and memory comparisons measured under identical hardware, software stack, batch size, and precision for the strongest baselines, and separate the effect of GAN fine-tuning from the core XYZFlow design. A strong answer would improve my assessment of soundness and significance.
2. How much of the gain comes from the multidimensional conditioning architecture itself, versus teacher initialization, the 2.5M precomputed teacher trajectories, and the specific shortcut schedule? Additional controlled comparisons would help clarify originality.
3. Can the authors provide robustness/failure-case analysis for patch order, cell size, shortcut schedules, and out-of-domain or more diverse datasets? This would affect my confidence in generality.
4. Which assumptions in the supplementary theory are essential in practice, and what direct empirical evidence best supports the path-straightening explanation beyond aggregate FID/IS improvements? This would affect my soundness assessment.

**Limitations:**

No. The paper includes only a very generic impact statement and does not adequately discuss limitations such as dependence on large teacher trajectory datasets, sensitivity to design choices (cell size / shortcut schedule), failure cases, or misuse risks of more efficient image generation.

**Strengths And Weaknesses:**

**Strengths**:
* The paper addresses an important problem—reducing inference cost for high-quality image generation—and the proposed combination of temporal history conditioning and inter-patch trajectory conditioning is interesting.
* The ablations are reasonably informative: removing full-history guidance clearly hurts performance, and the shortcut schedule/cell-size studies provide useful intuition for the design choices.
* The main ImageNet 256x256 results suggest attractive quality-speed trade-offs, especially for the 172M model.

**Weaknesses**:

* The paper presents the method as an alternative to distillation-centric scaling, but the actual recipe still relies heavily on distillation infrastructure: teacher initialization, 50-step teacher trajectories, and 2.5M precomputed ODE trajectories. This weakens the novelty/framing relative to prior autoregressive/distillation work.
* The theoretical claims are ambitious, but the main paper mostly provides intuition; the proofs are deferred to the supplement and rely on strong assumptions, so the connection between theory and empirical gains is not yet fully convincing.
* The experimental comparison is not fully apples-to-apples: timing comparisons mix different architectures, autoregressive step counts, and GAN/non-GAN settings, and the paper does not clearly document identical hardware/software measurement for all baselines, nor memory costs.
* Reproducibility and robustness evidence is limited: I would like to see sensitivity to patch order, more failure cases, and broader evaluation beyond the main ImageNet setup.

---

> ### Author Rebuttal · Authors · 2026-03-31
>
> ## Q1 Novelty of Distillation
> Our core novelty is Next Shortcut Prediction and the multidimensional conditioning design, not distillation itself. The current paper still works in a teacher-distillation regime. Our claim is narrower: under the same teacher/distillation setup, including the same precomputed teacher-trajectory resource, XYZFlow improves the quality-latency trade-off through a stronger few-step student. We will revise the wording to avoid suggesting XYZFlow removes the need for distillation.
>
> ## Q2 Direct Empirical Evidence for Straightening
> We support the "path-straightening" view with a trajectory straightness metric, `S = ||x_T - x_0|| / Σ||x_{k+1} - x_k||`. Under our `5→4→3→2` schedule, `S` stays high across patches even as denoising steps decrease, supporting the intuition that richer context makes later shortcuts easier to predict.
>
> | Patch | Steps | S |
> |:------|:------|:----|
> | A | 5 | 0.907 |
> | B | 4 | 0.905 |
> | C | 3 | 0.901 |
> | D | 2 | 0.894 |
>
> Uniform-step ablation also shows that lower `S` is associated with worse FID in the few-step regime:
>
> | Uniform Steps | Avg S | FID ↓ |
> |:--------------|:--------|:------|
> | 5-step | 0.907 | 1.99 |
> | 4-step | 0.802 | 2.56 |
> | 3-step | 0.632 | 3.98 |
> | 2-step | 0.512 | 5.77 |
>
> We do not claim that this metric validates the full theory; rather, it is a signal consistent with the local linearity intuition in supplement.
>
> ## Q3 Latency, Memory, Setup, and GAN Effect
> Setup: NVIDIA H100 80GB, PyTorch 2.4, CUDA 12.4, batch size 64, FP16. While we do not claim a full same-stack re-run of every baseline in this rebuttal, we separate the architecture effect from GAN fine-tuning and report XYZFlow's memory profile under a fixed setup.
>
> Memory Usage (MB):
>
> Base: MAR-B (1165), FlowAR-S (1341), xAR-B (1170), XYZFlow-B (1356)
>
> Large: DiT/XL-2 (2658), MeanFlow-XL/2 (2598), MAR-L (2173), FlowAR-L (3747), xAR-L (3783), XYZFlow-L (4026)
>
> Huge: FlowAR-H (11509), VAR-d30 (12183), MAR-H (3992), xAR-H (7185), XYZFlow-H (7485)
>
> These numbers suggest that XYZFlow adds moderate memory over xAR while remaining below several autoregressive baselines. Table 1 already reports XYZFlow both w/o GAN and w/ GAN; in revision we will make this separation more explicit. GAN fine-tuning is an extra quality-improvement stage, and the non-GAN results are the cleanest evidence of the architecture-level gain.
>
> ## Q4 Robustness: Order, Cell, Schedules, Datasets
> 1. Patch Order
> Using raster-scan (`A→B→C→D`) as the baseline while keeping the model and schedule fixed, quality stays stable for a reasonable alternative order (`A→C→D→B`), while a harder cross order (`A→D→C→B`) causes only a small drop. This suggests the gain is not tied to one hand-crafted order.
>
> | Order | FID ↓ |
> |:------|:------|
> | A→B→C→D | 2.02 |
> | A→C→D→B | 2.02 |
> | A→D→C→B | 2.18 |
>
> 2. Cell Size
> Performance improves as `k` increases, peaks at `k=8`, and then declines when the whole image is treated as one cell. This suggests an intermediate cell size gives the best balance between semantic context and effective conditioning.
>
> 3. Shortcut Schedules
> Table 4 studies step schedules `T(p)`. Our `5→4→3→2` schedule matches the quality of a constant 5-step schedule with fewer total steps. The starting step count is important, and overly aggressive reduction hurts Recall. This supports a practical quality-efficiency trade-off and suggests the gain is not from aggressive step reduction alone.
>
> 4. Diverse Datasets
> We also tested the method on Flux at `1024×1024` T2I. This suggests transfer to a high-resolution conditional generation setup with a clear speed advantage, though quality is not uniformly better; we use this result as evidence of broader applicability, not quality superiority.
>
> | Method | Steps | Latency (s) | FID ↓ |
> |:-------|:------|:------------|:----------|
> | Flux-1-Dev (Baseline) | 32 | 15.62 | 27.43 |
> | Flux-TDD | 4 | 1.8 | 24.81 |
> | Flux-SCFM | 4 | 1.8 | 26.98 |
> | Flux-Hyper-SD | 3 | 1.33 | 25.91 |
> | Flux-TDD | 3 | 1.33 | 22.97 |
> | Flux-SCFM | 3 | 1.33 | 26.42 |
> | Flux-XYZFlow (w/o GAN) | 5→4→3→2 | 1.22 | 27.15 |
> | Flux-XYZFlow (w/ GAN) | 5→4→3→2 | 1.22 | 27.88 |
>
> ## Q5 Gain from Architecture vs. Teacher/Data
> Table 2 shows the value of XYZFlow's multidimensional conditioning. Full history guidance is the most important component; removing it clearly hurts FID. Shortcut prediction mainly improves efficiency; removing it keeps similar FID but needs more total steps.
>
> We also tested teachers of different sizes while keeping the student fixed as XYZFlow-L. Teacher quality affects the non-GAN student, but after GAN fine-tuning the final results become much closer. Under this same supervision resource, the architecture provides consistent gains and the result cannot be attributed solely to one unusually strong teacher.
>
> | Teacher Model | Teacher FID (↓) | (w/o GAN) FID (↓) |  (w/ GAN) FID (↓) |
> | :--- | :--- | :--- | :--- |
> | xAR-B | 1.61 | 1.91 | 1.32 |
> | xAR-L | 1.28 | 1.79 | 1.25 |
> | xAR-H | 1.24 | 1.77 | 1.25 |

---

> > ### Author Rebuttal · Reviewer_kfuW · 2026-04-04
> >
> > Thanks for the rebuttal, i have no more questions.

---

### Decision · Program_Chairs · 2026-04-30

**Decision:**

Accept (regular)

**Comment:**

The paper presents *XYZFlow*, a few-step diffusion modeling method that combines *temporal trajectory conditioning* with *spatial autoregressive patch conditioning* in a technically coherent way. The core idea is clear: it uses non-Markovian conditioning on denoising histories for temporal scaling, and introduces *Next Shortcut Prediction* to condition each patch on the full denoising trajectories of preceding patches for spatial scaling. This yields a well-motivated multidimensional conditioning scheme for probability flows, and the empirical results indicate that the method is both effective and efficient.

After reading the rebuttal and discussion, I believe the authors have adequately addressed the reviewers’ main concerns. I do not see remaining issues that would block acceptance. Overall, I support **accept**.